

# The fate of fixed nitrogen in oligotrophic marine sediments: an in situ study

Stefano Bonaglia[1,†], Astrid Hylén[2], Jayne E. Rattray[1], Mikhail Y. Kononets[2], Nils Ekeroth[2,3], Per Roos[4], Bo Thamdrup[5], Volker Brüchert[1], Per O. J. Hall[2]

[1]Department of Geological Sciences and Bolin Centre for Climate Research, Stockholm University, Stockholm, Sweden
[2]Department of Marine Sciences, University of Gothenburg, Gothenburg, Sweden
[3]Calluna AB, Nacka, Sweden
[4]Center for Nuclear Technologies, Technical University of Denmark, Roskilde, Denmark
[5]Department of Biology and Nordic Center for Earth Evolution, University of Southern Denmark, Odense M, Denmark
[†]Current address: Department of Geology, Lund University, Lund, Sweden

*Correspondence to*: Stefano Bonaglia (stefano.bonaglia@gmail.com)

**Abstract.** Given the increasing impacts of human activities on global nitrogen (N) cycle, investigations on N transformation processes in the marine environment have drastically increased in the last years. Benthic N cycling has mainly been studied in anthropogenically impacted estuaries and coasts, while its understanding in oligotrophic systems is still scarce. Here we report on rates of denitrification, anammox and dissimilatory nitrate reduction to ammonium (DNRA) studied by in situ incubations with benthic chamber landers during two cruises to the Gulf of Bothnia (GOB), a cold, oligotrophic basin located in the northern part of the Baltic Sea. Burial and benthic solute fluxes were also experimentally determined to investigate the fate of fixed N in these sediments. Average rates of $N_2$ production by denitrification and anammox (range 53–360 µmol N m$^{-2}$ d$^{-1}$) were comparable to those from Arctic and subarctic sediments worldwide (range 34–344 µmol N m$^{-2}$ d$^{-1}$). Anammox accounted for 18–26 % of the total $N_2$ production. Absence of free hydrogen sulfide and low concentrations of dissolved iron in sediment pore waters suggested that denitrification and DNRA were driven by organic matter oxidation rather than chemolithotrophy. DNRA was as important as denitrification at a shallow, coastal station situated in the northern Bothnian Bay. At this pristine and fully oxygenated site, ammonium regeneration through DNRA contributed more than one third to the total dissolved nitrogen (TDN) diffusing from the sediment to the water column, and accounted, on average, for 45 % of total nitrate reduction. At the offshore stations, the proportion of DNRA in relation to denitrification was lower (0–16 % of total nitrate reduction). Median value and range of benthic DNRA rates from the GOB were comparable to those from the southern and central eutrophic Baltic Sea and other temperate estuaries and coasts in Europe. Therefore, our results contrast with the view that DNRA is negligible in cold and well-oxygenated sediments with low organic carbon loads. However, the mechanisms behind the variability in DNRA rates between our sites were not resolved. The GOB sediments were a major source (237 kt y$^{-1}$, which corresponds to 184 % of the external N load) of fixed N to the water column through recycling mechanisms. To our knowledge, our study is the first to document the simultaneous contribution of denitrification, DNRA, anammox and TDN recycling combined with in situ measurements.





## 1 Introduction

Excess of fixed N accumulating in aquatic ecosystems due to planktonic $N_2$ fixation, discharge of wastewater and runoff of fertilizers, constitutes one of the greatest eutrophication issues in coastal waters (Howarth and Marino, 2006), ultimately leading to water column anoxia and biodiversity loss (Vaquer-Sunyer and Duarte, 2008). Coastal, estuarine and shelf sediments contribute to ~45 % of the global N loss due to the two anaerobic, microbial processes denitrification and anammox (Seitzinger et al., 2006). Denitrification is the stepwise reduction of nitrate to nitrous oxide and dinitrogen gas ($N_2$) while anammox produces $N_2$ through ammonium oxidation coupled to nitrite reduction. These two processes help in counteracting eutrophication by permanent removal of fixed N from the system. A third N reducing process, DNRA, leads to recycling and preservation of fixed N in the system, and can ultimately increase the occurrence of algal blooms and exacerbate eutrophication if stimulated at the expense of the $N_2$-producing pathways (An and Gardner, 2002; Bonaglia et al., 2014a).

In sediments with high sulfide concentrations, DNRA is generally of major importance (Bonaglia et al., 2014a; Christensen et al., 2000; De Brabandere et al., 2015). This is likely because sulfide is used as electron donor in the nitrate reduction process which is carried out by large sulfur bacteria that proliferate in these conditions (Jørgensen and Nelson, 2004). DNRA dominates nitrate reduction also in sediments with high organic carbon (C) loading in tropical (Dong et al., 2011) and subtropical (An and Gardner, 2002) estuarine sediments, where high C:N ratios would favor this process over denitrification and anammox (Burgin and Hamilton, 2007; Kraft et al., 2014). It has recently been proposed that reduced iron can serve as an alternative electron donor for DNRA in estuarine sediments (Robertson et al., 2016). The few experimental studies conducted in permanently cold (< 10 °C), oligotrophic marine systems have suggested that the role of DNRA is negligible, while denitrification and anammox have been considered the main nitrate/nitrite (as $NO_3^- + NO_2^- = NO_x^-$) reduction processes (NRPs) (Crowe et al., 2012; Gihring et al., 2010). However, bacterial assimilation has also been found to be an important nitrate removal pathway in cold, oligotrophic Arctic sediments (Blackburn et al., 1996). In marine systems, anammox is, in relation to denitrification, generally more important in deep environments particularly in manganese (Mn)-rich sediments (Engström et al., 2005; Trimmer et al., 2013). Yet, factors responsible for the relative partitioning between the three NRPs in oligotrophic sediments are still obscure.

The southern and central Baltic Sea is one of the most eutrophic marine areas in the world, due to large inputs of nutrients, extended thermohaline stratification and limited water circulation (Elmgren, 2001). However, the northern part of the Baltic Sea — the GOB — is still relatively unaffected by anthropogenic nutrient loading because of low population density and extensive forest coverage in its catchment area (Pettersson et al., 1997), which prevent the risk of dense planktonic blooms similarly to the Arctic coastal zones (Billen et al., 2011). Planktonic primary production in the waters of the GOB is considered to be mainly limited by phosphorus (P), while dissolved inorganic N is generally abundant (Rolff and Elfwing, 2015; Tamminen and Andersen, 2007). In contrast to the southern Baltic, the GOB water column is well-oxygenated



suggesting that anaerobic NRPs happen exclusively in the sediments. To our knowledge, only two studies report rates of benthic denitrification in the GOB, but do not describe anammox and DNRA activity (Stockenberg and Johnstone, 1997; Tuominen et al., 1998).

This study targeted rates of NRPs in sediments of the GOB by in situ benthic lander incubations supported by additional on-deck incubations, as the main aim was to better understand processes of the N cycling in this relatively unexplored Baltic Sea basin. The oligotrophic, cold waters of the GOB make this basin an ideal environment to study the relative importance of individual NRPs under pristine conditions. Based on reported chemical data, we hypothesized low benthic N cycling rates dominated by denitrification and anammox, with the latter process prevailing at high Mn concentrations. A wide array of environmental parameters — including pore water chemistry, anammox biomarkers and C:N ratios — was characterized to assess their role in controlling N cycling processes. To investigate the fate of fixed N, N burial and efflux from the sediment were also quantified.

## 2 Methods

### 2.1 Study area and sampling

The GOB, which by area and volume represents the second largest basin of the Baltic Sea after the Baltic Proper, is the northernmost Baltic basin and is further divided into the Bothnian Bay (BB) and the Bothnian Sea (BS) (Fig. 1). The areas of the BB and BS cover 36,260 km$^2$ and 64,886 km$^2$, with mean depths of 41 and 66 m, respectively (Leppäranta and Myrberg, 2009). Both basins are normally covered by ice during winter for on average 120 and 60 days in the BB and BS, respectively (Håkansson et al., 1996). Bottom water salinity decreases from 6 in the southern section of the BS to 2 in the northern section of the BB. Due to sills and archipelago areas in the south of the BS, the GOB remains largely isolated from the density-stratified waters of the Baltic Proper. As such, and because of the low productivity and weak stratification of its water masses, the GOB is generally well-oxygenated throughout the year, and hypoxia has not significantly affected the GOB in the last centuries (Savchuk, 2013). The entire BB and the offshore waters of the BS are considered oligotrophic and P-limited in their current state (Billen et al., 2011; Tamminen and Andersen, 2007).

The four sampling stations (RA2, GOB1, GOB2, and GOB3) are located along a bottom water salinity gradient, on a north-south transect across the GOB (Fig. 1). RA2 is a shallow coastal station situated just outside the mouth of the Råne River, while GOB1, GOB2 and GOB3 are offshore stations. RA2, GOB1 and GOB2 are in the BB, while GOB3 is in the BS.

The stations were visited during two research expeditions, in June 2013 and in July 2014, with the only exception of station RA2, which was visited only during the second expedition (Table 1). In both years, in situ benthic chamber incubations were performed and sediment samples were collected for determination of various parameters. In 2014, we also performed on-deck incubations of sediment cores and anoxic slurries.





## 2.2 Sediment properties

For analysis of sediment physico-chemical properties and porewater, seafloors were sampled by means of a modified box corer (28 x 28 cm internal diameter) (Blomqvist et al., 2015) and by a Gemini corer (9 cm internal diameter). Both samplers provided nearly undisturbed sediment surfaces.

At each station, one Gemini core was sliced at intervals of 0.5 cm down to 2 cm depth; at intervals of 1 cm from 2 to 6 cm depth; and at intervals of 2 cm from 6 to 20 cm depth. Each sediment slice was split and one half frozen for organic geochemistry parameters, and the other refrigerated for later determination of water content and sediment accumulation rate (see below for analysis details). A second Gemini core was sliced at the same intervals to obtain porewater nutrient profiles. The sediment slices were centrifuged at 670 g (2500–3000 rpm) for 15 min and the supernatant was immediately filtered

using 0.45 µm polyethersulfone (PES) filters then stored dark and refrigerated until analysis. In 2013, $NH_4^+$ and the sum of $NO_2^- + NO_3^-$ ($NO_x^-$) were determined, while in 2014 the three dissolved inorganic nitrogen (DIN) species ($NH_4^+$, $NO_3^-$ and $NO_2^-$) and TDN were analyzed (see below for details).

Two small plastic liners (4.6 cm internal diameter) were inserted into the box core sample and sub-sampled for on deck microelectrode profiling of dissolved oxygen and sulfide concentrations. At least three to five microprofiles were measured

in each of the two sediment cores using a Clark-type oxygen microsensor (OX-50, Unisense) and a sulfide microsensor ($H_2S$-50, Unisense) mounted onto a double headed motorized micromanipulator (MM33-2, Unisense), using a vertical resolution of 100 µm. Sulfide microprofiles were carried out down to 5 cm depth, while the $O_2$ microprofiles were stopped immediately below the depth where $O_2$ was exhausted. An overlying water column of 4 cm was left in the sediment core and circulated by a gentle flow of air towards the water surface from an angle of ~45 ° in order to obtain a stable diffusive

boundary layer during measurements. Before measurements, OX-50 was calibrated using a 2-point calibration procedure, while $H_2S$-50 was calibrated daily in fresh anoxic $Na_2S$ solutions according to the manufacturer's recommendation.

## 2.3 In situ benthic chamber incubations

The two Gothenburg benthic landers (big and small landers) were deployed at each station to measure benthic solute fluxes and N cycling process rates (Brunnegård et al., 2004; De Brabandere et al., 2015; Ståhl et al., 2004). The big and small

landers are equipped with four and two box-shaped (20 cm x 20 cm) incubation chamber modules, respectively. The landers' chambers enclosed sediment together with the overlying water, which was constantly stirred by a horizontal paddle wheel positioned centrally in each chamber (Tengberg et al., 2004). The chambers' lids were closed 2.5–3 h after the benthic lander was deployed on the sediments to assure proper ventilation before incubation started. Physico-chemical conditions inside each chamber, as well as in the ambient bottom water just outside chambers, were monitored with an oxygen optode (3830

or 3835, Aanderaa) and a salinity sensor (3919A, Aanderaa). Both optodes and conductivity sensors had temperature output. Each chamber was equipped with ten 60 mL syringes for solution injection and water sampling. Three of the six chambers



were incubated for $O_2$ and TDN flux determination (Ståhl et al., 2004). In these chambers, 10 min after lid closed, the first syringe injected 60 mL distilled water, corresponding to 0.5–1 % of the chamber volume. Chamber volumes were calculated from the resulting decrease in salinity. Nine water samples were collected by syringe withdrawal at regular intervals during an incubation period of 29.5–33 h after the lid closed. After recovery of the landers, the syringe samples were filtered (0.45

µm, PES) and stored refrigerated until analysis for TDN concentrations immediately after each cruise.

The rest of the chambers were used for incubation with $^{15}NO_3^-$ tracer for determination of denitrification and DNRA rates following the protocol by De Brabandere et al. (2015), with minor modification. In each of these chambers: 10 min after the lid closed a syringe withdrew a first water sample for nutrient analyses. Ten min later, 60 mL of a 12 mM $^{15}NO_3^-$ solution (prepared by dissolving Na$^{15}NO_3$ 99.4 atom % (Sigma–Aldrich) in distilled water) was injected by the second syringe to

reach a final $^{15}NO_3^-$ concentration of ~70 µM in the chamber. After another 10 min the third syringe withdrew a second water sample for nutrient analysis, from which $^{15}NO_3^-$ amendment could be calculated. Seven water samples were collected by syringe withdrawal at regular intervals during an incubation period of 29–32.5 h after the initial operations were concluded. After recovery of the landers, the seven syringe samples from each of the $^{15}NO_3^-$-amended chambers were sampled first by filling a series of 12 mL Exetainers (Labco) to which 100 µL of a 37 % formaldehyde solution was added to

stop biological activity. The Exetainers were stored upside down in a fridge for later analysis of the isotopic composition of $N_2$. From the same syringe samples a second aliquot was filtered (PES, 0.45 µm) and split into two plastic vials, one for analysis of DIN, and one for $^{15}NH_4^+$ analyses. The nutrient vials were stored dark and refrigerated until analysis immediately after each cruise, while the $^{15}NH_4^+$ vials were immediately frozen upright.

**2.4 Sediment core incubations**

In 2014, two sediment box core casts were sampled by inserting 15 plastic liners (4.6 cm internal diameter, 30 cm length) and collecting half sediment and half water for sediment core incubations with the addition of $^{15}NO_3^-$ to determine rates of denitrification and DNRA (De Brabandere et al., 2015; Nielsen, 1992). These incubations were used to determine the relative accumulation of $^{15}N_2$ and $^{15}NH_4^+$ in the sediment and the overlying water, respectively, in order to correct rates from benthic chamber incubations, as this procedure does not take into account the $^{15}N_2$ and $^{15}NH_4^+$ fractions trapped in the

sediment (Nielsen, 1992). Since large chambers better capture the spatial heterogeneity of the sediment (Glud and Blackburn, 2002), we use the chamber-based rates as the best estimates of activity in situ.

The sediment cores were transferred into a 25 L incubation tank that was previously filled with ambient bottom water, situated in a temperature-controlled room kept at bottom water temperature (Table 1). The cores were left uncapped for 6 h, during which the water phase of each core was stirred with a magnetic bar driven by an external magnet at 60 rpm.

Subsequently, 5 mL of a 200 mM $^{15}NO_3^-$ solution (prepared by dissolving Na$^{15}NO_3$ 99.4 atom % (Sigma–Aldrich) in distilled water) was added to the water tank in order to reach a $^{15}NO_3^-$ concentration of ~70 µM. Before and after this




addition of $^{15}NO_3^-$ solution, triplicate water samples were collected from the tank, filtered (PES, 0.45 µm) and stored dark and refrigerated for later $NO_3^-$ analysis in order to calculate the final $^{15}NO_3^-$ amendment. The incubation started after a lag time of up to 14 h, which was necessary to homogeneously mix the added nitrate with the endogenous nitrate and to establish a linear production of $^{15}N_2$ within the sediment as inferred from the oxygen penetration depth (Dalsgaard et al., 2000). At the

beginning of the incubation the cores were capped with rubber stoppers so that no air bubbles formed and the water was mixed by externally driven magnetic bars. Triplicate cores were sampled at regular intervals during the incubation, which lasted 12 h at RA2 and 22 h at GOB1, GOB2 and GOB3. The $O_2$ concentration, monitored in a control core with an optode (3830, Aanderaa), did not decrease by more than 20 % of the initial value during the incubation time. The incubation was terminated by uncapping each core and sampling its water phase with a syringe. An aliquot was transferred into a 12 mL

Exetainer to which 100 µL of a 37 % formaldehyde solution was added. A second aliquot was filtered (PES, 0.45 µm), placed into a plastic vial, and immediately frozen for $^{15}NH_4^+$ analysis. Subsequently, the water phase and upper 7–9 cm of sediment were blended into slurry. Slurry samples were collected in 12 mL Exetainers to which 200 µL of a 37 % formaldehyde solution was added. The Exetainers were stored upside down in a fridge until later analysis of $N_2$ isotopic compositions. An additional sample of the slurry was taken from each core, centrifuged (670 g for 10 min), filtered (PES,

0.45 µm), placed into a plastic vial and immediately frozen for $^{15}NH_4^+$ analysis.

## 2.5 Anoxic slurry incubations

Anoxic slurry incubations amended with $^{15}NO_3^-$ and $^{15}NH_4^+$ were performed in order to estimate the contribution of anammox to total $N_2$ production during the 2014 expedition (Risgaard-Petersen et al., 2003; Thamdrup and Dalsgaard, 2002). The experiment followed the procedure described in Bonaglia et al. (2014b). Briefly, the oxic layers were removed

from two Gemini cores and the 2-cm thick sediment layers below were extruded and homogenized in a glass bottle filled with helium (He). 100 ml of this sediment was transferred to a second glass bottle filled with 900 mL filtered (PES, 0.45 µm), anoxic bottom water. This slurry was bubbled with He for 10 min to remove any oxygen that entered during previous operations, and was dispensed through a Viton™ tubing into a series of 12 mL Exetainers, each containing a 4 mm glass bead. The bottle was shaken vigorously while filling the Exetainers maintaining a homogeneous slurry throughout

dispensing. The Exetainers (n=36) were filled completely and directly capped to avoid air bubbles. The samples were pre-incubated for up to 16 h on a rotating stirrer in order to remove any residual oxygen and nitrate. After pre-incubation, 15 Exetainers received 100 µL of an anoxic 9 mM $^{15}NO_3^-$ solution (final $^{15}NO_3^-$ concentration: ~75 µM); 15 Exetainers received 100 µL of an anoxic 9 mM $^{15}NH_4^+$ + $^{14}NO_3^-$ solution (final $^{15}NH_4^+$ and $^{14}NO_3^-$ concentrations: ~75 µM); and 6 Exetainers received no tracer and were used as control. Triplicate vials from each treatment (n=9) were sampled directly with a syringe

by inserting a 3 mL He headspace. The subsample was centrifuged (670 g for 10 min), filtered (PES, 0.45 µm), placed into a plastic vial and stored cold for later $NO_3^-$ and $NH_4^+$ analysis, from which the label percentage of the $^{15}N$-compunds could be calculated. Formaldehyde (200 µL, 37 % solution) was injected into each of the He headspace Exetainers and mixed. The rest of the samples (n=27) were incubated on the rotating stirrer for up to 8 h at station RA2 and up to 18 h at stations GOB1,



GOB2 and GOB3. Triplicate vials from the $^{15}$N treatments were sacrificed at regular intervals during incubation by injecting 100 µL of 37 % formaldehyde. At the last time point, triplicate vials from the control treatment were also sacrificed to verify that no $^{15}$N-tracer contamination had occurred. All Exetainers were stored upside down in a fridge until analysis of isotopic compositions of N$_2$.

5 **2.6 Laboratory analysis and rate calculations**

Concentrations of TDN, NH$_4^+$, NO$_3^-$ and NO$_2^-$ were determined colorimetrically on a segmented flow nutrient analyzer system (OI Analytical, Flow Solution IV). The dissolved organic nitrogen (DON) concentration was calculated as the concentration of TDN minus the concentrations of NH$_4^+$ and NO$_x^-$. Benthic fluxes from lander incubations were calculated as the net concentration change of solutes (O$_2$ and TDN) per area and time (Ståhl et al., 2004). The concentrations were 10 corrected for the dilution that occurs when ambient bottom water enters the chamber as samples are withdrawn from the incubated chamber water. Benthic fluxes were calculated by multiplying the slope value of the regression of the concentration values versus time with the height of the incubated water column, which was estimated from the chamber volume. Fluxes were evaluated by the protocol described in Ekeroth et al. (2016), and considered to be significant if the $p$-value of the linear regression was $\leq 0.05$.

15 The isotopic composition of the N$_2$ samples from the denitrification and anammox experiments were determined by headspace analysis using gas chromatography-isotope ratio mass spectrometry (GC-IRMS, DeltaV plus, Thermo) (De Brabandere et al., 2015). Slopes of the linear regression of $^{29}$N$_2$ and $^{30}$N$_2$ concentration against time were used to calculate production rates of labeled N$_2$ ($p^{29}$N$_2$ and $p^{30}$N$_2$, respectively). Rates of N$_2$ production in benthic lander incubations ($p_{14}lan$), and in the water column ($p_{14}wc$) and slurried phase ($p_{14}sl$) of sediment core incubations were calculated based on the revised-20 isotope pairing technique (r-IPT) by Risgaard-Petersen et al. (2003). As $p_{14}lan$ takes into account only the N$_2$ diffusing to the water column, the in situ N$_2$ production rate ($p_{14}$) was calculated using the following formula:

$$p_{14} = p_{14}lan \, / \, F_{wc} \qquad (1)$$

where $F_{wc}$ is the fraction of N$_2$ production diffusing to the water column of sediment core incubations calculated as:

$$F_{wc} = p_{14}wc \, / \, p_{14}sl \qquad (2)$$

25 Sediment core incubations were only carried out in 2014, but because sediment geometries and zonation were similar in the two years we expected the same $F_{wc}$, consistent with De Brabandere et al. (2015). The in situ N$_2$ production rate was partitioned into N$_2$ production coupled to nitrification ($p_{14}n$) and N$_2$ production depending on the water column nitrate ($p_{14}w$) according to Risgaard-Petersen et al. (2003).



The contribution of anammox ($ra$) to the total $N_2$ production — necessary to calculate $p_{14}$ (Risgaard-Petersen et al., 2003) — was estimated from anoxic slurry incubations. As significant rates of DNRA were measured in benthic lander incubations (see below), a correction to account for $^{15}NH_4^+$ production in Exetainers incubated with $^{15}NO_3^-$ was performed based on Song et al. (2013). $F_A$, the fraction of $^{15}NH_4^+$ in the ammonium pool in the nitrate reduction zone, was estimated from the concentrations of $^{15}NH_4^+$ and $^{14}NH_4^+$ at the final time point (tenth syringe) from benthic lander incubations. $F_A$ in these incubations was $\leq 0.06$. The in situ anammox rate (AAO) was calculated by multiplying $p_{14}$ by $ra$. The in situ denitrification rate (DEN) was calculated as the $p_{14}$ minus AAO.

Concentrations of labeled ammonium ($^{15}NH_4^+$) were quantified after oxidation of $NH_4^+$ to $N_2$ with hypobromite (Warembourg, 1993). Samples were purged with helium for 10 min, treated with alkaline hypobromite, and analyzed by the headspace technique as described above. Slopes of the linear regression of $^{15}NH_4^+$ concentration against time were used to calculate production rates of labeled ammonium ($p^{15}NH_4^+$) after correction for adsorption assuming an adsorption coefficient of 1 (De Brabandere et al., 2015). DNRA rates in benthic lander incubations (DNRA$lan$) were calculated according to Christensen et al. (2000). Similar to $N_2$ production rates, the in situ DNRA rate was calculated from the formula:

$$\text{DNRA} = \text{DNRA}lan\ /\ F_{wc} \tag{3}$$

where $F_{wc}$ was used because $p^{15}NH_4^+$ was not detected in the time courses from sediment core incubations at GOB1–3 (see Results).

Sediment samples used for water content determination were dried at 70 °C for 4–9 days until reaching a constant weight. Porosity was calculated from the water content assuming a dry sediment density of 2.65 g mL$^{-1}$. The dried sediment was ground into a homogeneous powder and was analyzed for total carbon (TC) and total nitrogen (TN), as well as organic carbon ($C_{org}$) and organic nitrogen ($N_{org}$) after treatment with HCl fumes, with a Carlo ERBA N1500 gas chromatograph (Verardo et al., 1990).

Frozen sediment samples used for the analysis of ladderane lipids and the sediment accumulation rate (SAR) were weighed, freeze-dried and weighed again. The freeze-dried sediment was manually homogenized into fine powder. Gamma emitting radioisotopes for $^{210}Pb$ dating were analyzed based on Cutshall et al. (1983). Briefly, 15 mL of sediment powder was packed into a Petri dish plastic container, weighed and placed on a high purity germanium (Ge) detector with a beryllium window for 1–2 days. For the surface sediment samples with a restricted dry sample mass a well-type Ge detector was used. After the counting period, sample self-absorptions were analyzed according to Cutshall et al. (1983). The $^{210}Pb$ excess activities in the samples were obtained by subtracting the $^{226}Ra$ activity from the gross $^{210}Pb$ activity. For each sediment profile, the excess $^{210}Pb$ activity concentration (Bq kg$^{-1}$) was plotted versus the cumulative sediment mass (g cm$^{-2}$) centered at the middle of each sediment layer. As bioturbation was absent (RA2), or low and restricted to the top sediment cm (GOB1–3), an



exponential regression curve of the form $e^{-kx}$ was fitted to the data, where x is the cumulative sediment depth (g cm$^{-2}$) and k is a constant. Using the half-life of $^{210}$Pb (22.4 years), SAR (g cm$^{-2}$ y$^{-1}$) was determined from (ln(2)/22.4)/k. The N burial rate was calculated for each station by multiplying the SAR with the average sedimentary N content at the depth of 14–20 cm, i.e. where it had reached a stable value.

For ladderane lipid analysis (anammox biomarker), 3 g of ground sediment was extracted using a modified method (additional extraction step using dichloromethane (DCM)) of Matyash et al. (2008). Extracts were combined to produce a total lipid extract which was subsequently dried under N$_2$. Directly prior to analysis the total extract was dissolved in 9:1 Methanol (MeOH):DCM (Zhu et al., 2013) and filtered using a 0.45µm pore size, 4 mm diameter polytetrafluoroethylene (PTFE) syringe filter. The target biomarker, a C$_{20}$-[3]-ladderane monoether attached to a phosphocholine (PC) head group,

was analyzed using high performance liquid chromatography electrospray ionization tandem mass spectrometry (HPLC-ESI-MS/MS, Thermo). Separation was performed using reverse phase chromatography on a Kinetex XB C18, 1.7µm, 100Å column (Phenomenex, USA) based on the methods of Lanekoff and Karlsson (2010) and Zhu et al. (2013). The amount of PC-C$_{20}$-[3]-ladderane monoether in the samples was quantified using a PC-C$_{20}$-[3]-ladderane monoether standard, purified from anammox cell biomass from the anammox waste water treatment reactor at Syvab, Himmerfjärdsverket, Grödinge,

Sweden. The standard was purified by adapting the semi-preparative HPLC method of Jaeschke et al. (2009) to reverse phase. Standard purity was determined by measuring the P content by spectrophotometry (Evolution 260, Thermo) and comparing this to the calculated P molar weight of the standard (Jaeschke et al., 2009).

**2.7 Mass balance calculations and data analysis**

A benthic mass balance was calculated to assess whether the sediments were sources or sinks of N to/from the water column.

It was calculated assuming steady state conditions where N deposition on the sediment surface equals the sum of burial, N$_2$ production, and the TDN efflux from the sediment. At each station, average rates of burial, N$_2$ production and TDN exchange were calculated for 2013 and 2014, and the average biannual values were summed to calculate N deposition. The relative contribution (%) of each of the three processes was calculated by dividing each process by N deposition.

If not stated otherwise in the text, measurements are reported as average ± standard error (SE). Detection limits of in situ

rates were estimated as the median value of 2 × SE of the significant rates (De Brabandere et al., 2015), and were 3.4 µmol N m$^{-2}$ d$^{-1}$ for DNRA and 29 µmol N m$^{-2}$ d$^{-1}$ for $p_{14}$. Statistical tests were performed in order to detect differences in solute fluxes and process rates between stations. Homogeneity of variance of the dataset was checked using Cochran's test. One way analysis of variance tests were performed. When the differences in the mean values among stations were greater than would be expected by chance, pairwise multiple comparisons among stations were performed by the Tukey test. Correlations

between process rates and environmental factors were tested using the Pearson's Correlation Coefficient ($r$). The level of



significance was always set to $p < 0.05$. Statistical analyses were performed with SigmaPlot 13.0 (Systat Software, CA, USA).

## 3 Results

### 3.1 Sediment properties and macrofauna

Sediment at GOB1 was dark olive brown in the top 3 cm, where a few burrows of the amphipod *Monoporeia affinis* were visible against the walls of the plastic liners. Sediment was light olive brown from 3 to 6 cm depth, and characterized by brown/black laminations below 6 cm depth. Sediment at GOB2 was similar to GOB1 but with grey/black laminations instead. GOB3 sediment was light olive brown until 6 cm depth, followed by a grey clay layer. Both GOB2 and GOB3 sediments presented minor burrow structures of *M. affinis* and of the polychaete *Marenzelleria* spp. down to 6-7 cm depth.

RA2 sediment was fluffy and olive brown-colored in the top cm, followed by a black/dark olive brown layer. No burrows could be found at RA2. Relative abundances of macrofaunal taxa in the GOB sediments are presented in Supplementary Fig. S1.

Sediment porosities were highest at RA2, followed by GOB2, GOB1 and GOB3, and depth-integrated porosities ranged 0.82–0.93 at the four sampling sites (Supplementary Fig. S2). Sedimentary $C_{org}$ and N generally decreased with sediment

depth, except for GOB3 in 2014 where they increased slightly below 2.5 cm depth (Fig. 2). At RA2, both $C_{org}$ and N were constant until 4.5 cm, below which they started decreasing. The $C_{org}$ and N content at the sediment surface (top 2 cm) were 4–5 % and 0.4–0.5 % dry weight, respectively, except at GOB3 where they were ~2.5 % and ~0.25 % dry weight. Sediment $C_{org}$:N molar ratios increased slightly with depth at all stations except at RA2, where they were relatively constant (Supplementary Fig. S3). Sediment surface $C_{org}$:N ratios were 12–14 at all stations, except of GOB3 where they were lower

(10–12).

Porewater nitrite ($NO_2^-$) concentrations were low at all stations ($< 0.8$ μM), while nitrate ($NO_3^-$) or the nitrate–nitrite sum ($NO_x^-$) showed a concentration peak (6-18 μM) between the sediment surface and down to 1.25 cm. Ammonium ($NH_4^+$) and dissolved organic nitrogen (DON) concentrations in the porewater increased with sediment depth at all stations (Fig. 3). Between the four stations, $NH_4^+$ concentrations decreased in the order RA2 > GOB2 ≥ GOB1 > GOB3. An $NH_4^+$ depleted

surface layer ($<1.5$ μM) was absent at RA2 but present and increasing in thickness from north to south at the GOB stations, ranging from 0.75 cm at GOB1 to 2.5 at GOB3. GOB2 had the highest DON concentrations, ranging from 406 to 677 μM below 5.5 cm depth. DON was the dominant form of total dissolved nitrogen (TDN) in the porewater at all stations except for RA2, where $NH_4^+$ was the dominant species in the TDN pool.

The sediment accumulation rates (SAR) at the four stations varied between 0.022 (GOB3) and 0.032 g cm$^{-2}$ y$^{-1}$ (GOB1)

(Table 1). Burial rates of N ranged from 0.098 (GOB1) to 0.15 mmol N m$^{-2}$ d$^{-1}$ (RA2) (Table 1). The oxygen penetration





depth (OPD) was higher in 2013 than in 2014 (Table 1). RA2 had lower OPD (0.24 cm) than the other three stations, where it ranged from 0.80 to 1.9 cm (Table 1).

The targeted ladderane lipid anammox biomarker (PC-$C_{20}$-[3]-ladderane monoether) could be detected at all stations at abundances that were highly variable, both spatially and temporally (Fig. 4). PC-$C_{20}$-[3]-ladderane monoethers were more

abundant in the top sediment layer and decreased with depth, with the exception of GOB3 in 2014, where a subsurface peak was recorded (Fig. 4). Higher abundances were present in 2013 compared to 2014 at GOB1 and GOB2, and the opposite was observed at GOB3. Average abundances in the top 4 cm of sediment were in the range 40–1336 pg g$^{-1}$ sediment dry weight (DW) equivalent to 5–220 pg g$^{-1}$ sediment wet weight (WW) (Table 2).

### 3.2 Benthic exchange of solutes and rates of N cycling processes

Total oxygen uptake (TOU) varied between -5.0 (GOB2, 2013) and -11.3 mmol $O_2$ m$^{-2}$ d$^{-1}$ (RA2, 2014) (Fig. 5a) and was significantly higher at RA2 than at the other stations (ANOVA, $p < 0.001$). Separate fluxes of $NH_4^+$ and $NO_x^-$ were mainly non-significant (data not shown) contrarily to the TDN fluxes, which were found to be significant and varied between 0.1 mmol N m$^{-2}$ d$^{-1}$ at GOB1 in 2014 and 0.9 mmol N m$^{-2}$ d$^{-1}$ at RA2 (Fig. 5b). The facts that the contribution of DIN to the TDN flux was minor, and that the porewater concentrations just below the sediment surface were dominated by DON,

clearly indicate that the TDN efflux was mainly supported by DON. TDN flux was non-significant at GOB3 in 2014. RA2 had a significantly higher TDN flux than GOB1 (2014) and GOB3 (ANOVA, Tukey pair-wise test, $p < 0.001$).

Sediment core incubations revealed that most of the $^{15}N_2$ produced remained trapped in the sediment during incubation, $F_{wc}$ being 0.26, 0.23, 0.33 and 0.21 at GOB1, GOB2, GOB3 and RA2, respectively. Anoxic slurries incubated with $^{15}NH_4^+$ and $^{14}NO_3^-$ revealed anammox activity in all the investigated sediments, with potential rates ranging from 0.2 to 0.8 nmol N g$^{-1}$

WW h$^{-1}$ at GOB1 and GOB3, respectively (Table 2). Potential denitrification rates (from $^{15}NO_3^-$-amended anoxic slurry incubations) were higher than anammox and varied from 1.0 to 3.5 nmol N g$^{-1}$ WW h$^{-1}$ at GOB1 and GOB3, respectively (Table 2).

Trends of in situ $N_2$ production (53–360 µmol N m$^{-2}$ d$^{-1}$), denitrification (43–297 µmol N m$^{-2}$ d$^{-1}$), and anammox (10–63 µmol N m$^{-2}$ d$^{-1}$), were similar between stations and rates were highest at RA2, and progressively lower at GOB2, GOB1 and

GOB3 (Fig. 6a, b, c). $N_2$ production was mainly sustained by $NO_3^-$ produced within the sediment by nitrification ($p_{14}n$), which contributed 92, 90, 87 and 83 % at GOB2, GOB1, RA2 and GOB3, respectively (Fig. 6a). Thus, $N_2$ production dependent on water column $NO_3^-$ ($p_{14}w$) was minimal. Rates of total $N_2$ production, $p_{14}n$, $p_{14}w$ and denitrification were significantly higher at RA2 (ANOVA, $p < 0.01$) than at the other stations. Anammox rates at RA2 and GOB2-2014 were significantly higher than at GOB3 (ANOVA, Tukey pair-wise test, $p < 0.01$). The contribution of anammox to total $N_2$

production ($ra$) was 18 % at RA2, 19 % at GOB1 and GOB3, and 26 % at GOB2 meaning that denitrification was more




important than anammox at all stations (Fig. 6b, c). There was no statistically significant difference (ANOVA, $p = 0.167$) in anammox contribution between different stations.

At GOB1–3, DNRA could not be determined in sediment core incubations because $p^{15}NH_4^+$ was not detectable in the time courses, and this is consistent with the higher detection limits of core incubations compared to lander incubations. Thus, the in situ DNRA rates were calculated using $F_{wc}$ values from $^{15}N_2$ accumulation in core incubations (see above), which may have caused an underestimation of the in situ DNRA rates because $NH_4^+$ diffusion to the water column is slower than $N_2$ diffusion. DNRA was not detectable in situ at GOB3, which is consistent with low to undetectable efflux of TDN measured in benthic chambers without $^{15}NO_3^-$ addition at this station (Fig. 5b, 6d). DNRA rates were lower than denitrification rates at GOB1 and GOB2 (ranging from 8 to 22 µmol N m$^{-2}$ d$^{-1}$) (Fig. 6d). Station RA2 displayed DNRA rates which were comparable to the denitrification rates, 266±90 and 297±49 µmol N m$^{-2}$ d$^{-1}$, respectively. RA2 had significantly higher DNRA rates than the other stations (ANOVA, p<0.001).

## 4 Discussion

### 4.1 Coexistence of multiple nitrate/nitrite reduction processes

We measured rates of $N_2$ production and DNRA by means of benthic chamber lander incubations, which have been shown to give the most reliable results of benthic transformation processes (Glud and Blackburn, 2002; Tengberg et al., 1995), and by correcting these rates with the relative anammox contribution determined in parallel anoxic slurry incubations (Risgaard-Petersen et al., 2003). The in situ incubations indicated that denitrification, DNRA and anammox all contributed significantly to NRPs in the GOB sediments. To our knowledge, our study is the first that detects simultaneous activity of denitrification, DNRA, and anammox with in situ measurements. The $^{15}N$ isotope pairing technique (IPT) may be inaccurate when all three NRPs coexist as in anaerobic slurry incubations DNRA reduces $^{15}NO_3^-$ to $^{15}NH_4^+$, and this may affect the isotope distributions of anammox and denitrification when DNRA activity is high and the background concentration of $^{14}NH_4^+$ is low (Song et al., 2016). At the stations where DNRA was detected, $F_A$ (the fraction of $^{15}NH_4^+$ in the ammonium pool) was 1.7 % at GOB1 and GOB2, and 5.6 % at RA2 which would lead to a simultaneous < 5 % overestimation of denitrification rate and underestimation of anammox rate (Song et al., 2016). Although this represents a small inaccuracy, we accounted for the influence of coupled DNRA-anammox during calculations, and our rates therefore represent in situ conditions.

Previous attempts to discern between the three NRPs in oligotrophic marine sediments involved the use of ex situ sediment core incubations (Crowe et al., 2012; Gihring et al., 2010). However, this method may not be representative of the in situ biogeochemical conditions when N cycling rates are low, e.g., at cold, low productivity sites. In Arctic sediments, DNRA was indeed not detected (< 50 µmol N m$^{-2}$ d$^{-1}$) (Gihring et al., 2010). Benthic chamber incubations have 2.7 and 4.5 fold lower detection limits for denitrification and DNRA rate measurements than sediment core incubations, respectively (De




Brabandere et al., 2015). This is due to core-to-core variation compared to repeated sampling of the same chamber. In addition, ex situ experiments generally involve heavy manipulation of the sediments, which can lead to alteration of sediment structures and infaunal biomass when small cores are used for sampling and incubation (Glud and Blackburn, 2002).

Rates of NRPs were dominated by denitrification (rate 43–297 µmol N m$^{-2}$ d$^{-1}$; median 91 µmol N m$^{-2}$ d$^{-1}$), which was in the lower range of rates previously measured in the GOB (0–940 µmol N m$^{-2}$ d$^{-1}$ (Stockenberg and Johnstone, 1997; Tuominen et al., 1998)). However, those previous studies could not take into account for potential anammox activity and may not be representative of the in situ rate (Risgaard-Petersen et al., 2003). Denitrification rates from this study were in the lower range of rates found in the adjacent Baltic Proper and Gulf of Finland (38–1619 µmol N m$^{-2}$ d$^{-1}$; median 170 µmol N m$^{-2}$ d$^{-1}$

(Bonaglia et al., 2014a; Deutsch et al., 2010; Hietanen and Kuparinen, 2008; Jäntti et al., 2011)). However, our rates were comparable to those reported from Arctic and subarctic sediments (33–340 µmol N m$^{-2}$ d$^{-1}$ (Gihring et al., 2010; Rysgaard et al., 2004; Seitzinger and Giblin, 1996)). Anammox rates (6–75 µmol N m$^{-2}$ d$^{-1}$; median 27 µmol N m$^{-2}$ d$^{-1}$) were higher than those measured in coastal areas of the Gulf of Finland and the central Baltic Proper (0–38 µmol N m$^{-2}$ d$^{-1}$; median 11 µmol N m$^{-2}$ d$^{-1}$ (Bonaglia et al., 2014a; Hietanen and Kuparinen, 2008)), but comparable to rates of Arctic sediments from Greenland and Svalbard (1–92 µmol N m$^{-2}$ d$^{-1}$ (Gihring et al., 2010; Rysgaard et al., 2004)) and those of Celtic and Irish Sea sediments

(2–46 µmol N m$^{-2}$ d$^{-1}$ (Jaeschke et al., 2009)). It thus seems that the GOB sediments behave similarly to Arctic and subarctic sediments in terms of global N loss and partitioning between denitrification and anammox. Alike those sediments, denitrification might be limited by the availability of organic matter, which might increase the contribution of anammox to N loss compared to sediments at lower latitude (Rysgaard et al., 2004).

DNRA rates found in this study (0–266 µmol N m$^{-2}$ d$^{-1}$; median 19 µmol N m$^{-2}$ d$^{-1}$) were in the lower range of rates reported in previous studies from anthropogenically impacted Baltic Sea estuaries and hypoxic shelf sediments (7–1060 µmol N m$^{-2}$ d$^{-1}$; median 23 µmol N m$^{-2}$ d$^{-1}$ (Bonaglia et al., 2014a; Jäntti et al., 2011)), albeit median values between these two environments were comparable. Interestingly, these comparisons suggest that the overall importance of DNRA in Baltic Sea sediments is similar across the gradient of trophic conditions. The only successful measurement of DNRA in Artic, subarctic

and boreal sediments to date describes rate of DNRA of only 0.12 µmol N m$^{-2}$ d$^{-1}$ in the Lower St. Lawrence Estuary (Crowe et al., 2012). Tropical and subtropical estuaries with high C$_{org}$ and nitrate concentrations, on the other hand, have been shown to be dominated by DNRA at rates up to 27,288 µmol N m$^{-2}$ d$^{-1}$ (An and Gardner, 2002; Dong et al., 2011). Temperate estuarine and coastal sediments investigated by the $^{15}$N-labeling technique have in situ rates that are intermediate between those extreme values (0–130 µmol N m$^{-2}$ d$^{-1}$ (Christensen et al., 2000; Rysgaard et al., 1996)), and are comparable to our

range.

While the presence of anammox bacteria was further substantiated by the presence of ladderane lipids, a biomarker proxy for presence of anammox bacteria (Jetten et al., 2009), at all stations, pooled ladderane abundances did not correlate with the



average anammox rates (Table 2) ($r = 0.47$, $p > 0.05$). In our study we specifically targeted the intact PC-monoether ladderane as it represents a more accurate proxy for living anammox cells than core ladderane lipids (Brandsma et al., 2011; Jaeschke et al., 2009). Our findings contrast with those of Bale et al. (2014), who observed a good agreement between intact PC-monoether ladderane abundance, potential anammox rates, 16S rRNA and hszA gene copy abundance in North Sea

sediment. However, other studies have also shown that the abundances of PC-monoether ladderane can be weak indicators of anammox activity (Jaeschke et al., 2009), possibly due to the fact that the PC-monoether ladderanes may be degradation lysis products of intact polar lipids (Brandsma et al., 2011). The abundance of PC-monoether ladderanes generally decreased with sediment depth indicating that living anammox bacteria were mainly distributed close to the sediment surface, as previously reported for shallow sediments off northwest Africa and in the Irish Sea (Jaeschke et al., 2010; Jaeschke et al.,

2009). The range of abundance of PC-monoether ladderanes found in this study (40–1336 pg g$^{-1}$) was up to two orders of magnitude higher than that from Celtic and Irish Sea sediments (0–60 pg g$^{-1}$ (Jaeschke et al., 2009)), and that from north-west African sediments (< 1–30 pg g$^{-1}$ (Jaeschke et al., 2010)), but comparable to the range reported in organic-rich, muddy sediments from the North Sea (100–1250 pg g$^{-1}$ (Lipsewers et al., 2014)).

## 4.2 Control factors of nitrate/nitrite reduction processes

In marine sediments, denitrification and DNRA are mainly driven by the oxidation of organic C and hydrogen sulfide (Canfield et al., 2005). In the sediments of the Gulf of Bothnia we consider a major dependence of denitrifiers and DNRA bacteria on hydrogen sulfide unlikely as this compound never accumulated in the porewater at any of the investigated stations. It has recently been shown that reduced dissolved iron ($Fe^{2+}$) can serve as electron donor for DNRA both in synthetic groundwater (Coby et al., 2011) and in seasonally hypoxic estuarine sediments (Robertson et al., 2016). In those

sediments, DNRA was most active in sites with high porewater $Fe^{2+}$ concentrations (>100 µM) in the top mm below the sediment surface (Robertson et al., 2016). We also consider this possibility unlikely for the sediments of the GOB as concentrations of $Fe^{2+}$ in the porewater of the top 2 cm of sediment were more than an order of magnitude lower than this, ranging from 3 to 12 µM (Hannah S. Weber, unpublished). In particular, RA2, the station with the highest rates of DNRA, had porewater $Fe^{2+}$ concentrations of only 5 µM in the top sediment layer.

The absence of dissolved sulfide and low concentrations of $Fe^{2+}$ in the GOB sediments rather indicate that the quantity and/or quality of organic matter exerted considerable influence on denitrification and DNRA, and suggest that the apparent electron donor for nitrate reduction was organic C and not an inorganic substrate. Rates of heterotrophic nitrate respiration were generally low because of the oligotrophic nature of this ecosystem, which provides low organic C loading to the sediments. Nitrate respiration by denitrification and DNRA only accounted for <1 % of the total C mineralization inferred

from the TOU at GOB3 and ~2 % at GOB1 and GOB2, which is comparable to estuarine Baltic Proper sediments where rates were limited by labile C supply (Bonaglia et al., 2014a). At station RA2 nitrate respiration made up 5 % of the total C mineralization, and this higher percentage is consistent with the lower OPD at this station than at the other stations (Table 1).



With similar $C_{org}$ and N contents among stations, the relatively higher contribution of nitrate respiration at RA2 might be explained by process dependence on organic C quality rather than quantity, because at the offshore stations high remineralization rates in the water column render the sinking material more refractory (Algesten et al., 2004).

In continental shelf sediments with high reactive Mn content, it has been shown that denitrification is outcompeted by Mn reduction as C remineralization process, which renders anammox relatively more important than denitrification (Thamdrup, 2012; Trimmer et al., 2013). In the Bothnian Bay there was a clear increase in surface sediment Mn content (dithionite-extractable) with water depth, being 47, 66 and 158 µmol $g^{-1}$ DW at RA2, GOB1 and GOB2, respectively (Hannah S. Weber, unpublished). However, anammox contribution and rates did not differ significantly between these three stations, indicating that the effect of Mn content on nitrate reduction in the GOB was much weaker than that reported from the deep Skagerrak, where denitrifiers were almost totally outcompeted by Mn-reducers at Mn contents of 270–421 µmol $g^{-1}$ (Trimmer et al., 2013). In benthic environments where phytoplanktonic organic matter *sensu* Redfield (C:N ≈ 6.6) is oxidized with nitrate as electron acceptor through NRPs, the predicted anammox contribution is 29 % (Thamdrup, 2012). However, C:N ratios in the GOB were always higher than Redfield's ratio, which may eventually explain the lower anammox contribution (18–26 %) to total $N_2$ production that we found in the GOB sediments.

DNRA contribution to total nitrate reduction ranged from 7 to 63 % in the subarctic, oligotrophic Bothnian Bay. Because of the conservative method we used to upscale the DNRA rates from the chamber incubations (see Methods and Results), these DNRA contributions are likely underestimates. Yet, to our knowledge, our study is the first that demonstrates significant, even predominant (in one lander's chamber at RA2) DNRA rates in an oligotrophic, cold benthic environment. These results are in contrast with what has been proposed in the recent literature, i.e., that DNRA is negligible in oligotrophic systems of cold and well-oxygenated waters (Crowe et al., 2012; Gihring et al., 2010). DNRA was of major significance at the shallow, oligotrophic site, where the relatively high C:N ratios of ~14 might provide a competitive advantage for DNRA bacteria vs. denitrifiers (Hardison et al., 2015; Kraft et al., 2014). Strains of the bacterium *Shewanella* spp., a genus that performs DNRA, were recently isolated in Arctic fjord sediments and their highest optimal growth rate was at 18 °C, while denitrifiers had their optima at 0 °C (Canion et al., 2013). We thus speculate that DNRA bacteria may have the capacity to increase their activity when the temperatures increased by ~6 °C from the winter to the summer, which was the case at RA2. It is clear from these results that further understanding of the controlling factors of heterotrophic DNRA in oligotrophic environments is necessary and such studies should focus on kinetic experiments coupled to molecular analyses of DNRA bacteria.

**4.3 Fixed N loss vs. recycling in Gulf of Bothnia sediments**

We estimated the N mass balance for the benthic GOB environment assuming that N deposition on the sediment surface equals the sum of burial, $N_2$ production and total dissolved N efflux from the sediment (see Methods for calculations). The relative contribution of the N turnover mechanisms was comparable between the investigated stations (Fig. 7). Only 36–46





% of the particulate organic N that sinks to the GOB sediments was permanently lost from the system. We estimated that bacterial $N_2$ production and N burial together removed on average 160 kt N $y^{-1}$ from the entire basin, which is 1.2-fold higher than the total external N load (Savchuk, 2005). The removal rate, as much as the recycling rate, was constrained by the low rates of N cycling processes at GOB3. The calculated flux of unsupported $^{210}Pb$ was substantially lower at GOB3 (59 Bq $m^{-2}$

$y^{-1}$) than at the other three stations (121–230 Bq $m^{-2}$ $y^{-1}$), which may indicate that this station temporally exhibited accumulation and temporally erosion behaviour. Unlike the other stations, there was no clear decrease of $C_{org}$ or N content down-core at GOB3, and the C:N ratios were relatively constant throughout the sediment profiles. Interestingly, in the 2014 profiles, both $C_{org}$ and N content increased slightly with depth at this station. C:N ratios generally increase when organic matter is decomposed due to preferential remineralization of N compared to C (Canfield et al., 2005), so these observations

may prove a decrease in organic matter degradation in the last few decades at this station.

On the other hand, the GOB sediments recycled 54–64 % of the deposited N to the water column in the form of TDN, which is close to the recycling percentage reported from eutrophic sediments, such as those in the shallow Yangtze Estuary (Deng et al., 2015) and the North Sea coast (Billen, 1978). Thus, it appears that in the summer the majority of the particulate organic N that sinks to the GOB sediments returns to the water column. The calculated basin-wise recycling rate of 237 kt N

$y^{-1}$ was 1.8-fold higher than the terrestrial N load plus the N input from atmosphere (Savchuk, 2005). This high internal N recycling into the benthic–pelagic system of the GOB sustains water column primary production and may be a contributing factor to the strong P limitation of this basin (Rolff and Elfwing, 2015). In particular, DNRA in the shallow coastal sediment of the Bothnian Bay sustains up to 37 % of the N recycled from the sediment to the water column, suggesting that this microbial process must be taken into account in N budgets not only in euxinic, but also in oligotrophic systems. Although the

assumption of steady state contains uncertainties as deposition and rates of N cycling may vary on a seasonal scale, our calculations give an indication of the relative importance of N loss vs. recycling in the GOB sediments during the period of maximal primary production (Klais et al., 2011).

**Acknowledgements**

We acknowledge support from the Swedish Research Council, VR (Grant no. 2012-3965 to P.H.), Stockholm University's

Baltic Ecosystem Adaptive Management, BEAM (funding to S.B. and V.B.), the Swedish Research Council for Environment, Agricultural Sciences and Spatial Planning, FORMAS (Grant no. 215-2009-813 to V.B.) and RA4 start-up grant from the Bolin Centre for Climate Research, Stockholm University (J.E.R.). We are grateful to the captains and crew of RV *KBV005* and RV *Fyrbyggaren*, and to Sarah Conrad and Susanne Bauer, for their assistance during work at sea. We thank the staff at the chemical lab at DEEP, Stockholm University, for assistance with nutrient analysis and Sara Söhr at

Syvab for the anammox biofilms.



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




**Tables**

**Table 1: Main site parameters obtained at the investigated sites. SAR is the sediment accumulation rate and OPD is the average oxygen penetration depth.**

| Station | Sampling season | Coordinates | Depth (m) | Temperature (°C) | Salinity | SAR (g cm$^{-2}$ y$^{-1}$) | Burial rate (mmol N m$^{-2}$ d$^{-1}$) | OPD (cm) |
|---------|-----------------|-------------|-----------|------------------|----------|----------------------------|------------------------------------------|----------|
| GOB1 | June 2013 | 23°23.7' E, 65°11.5' N | 86 | 2.6 | 3.5 | 0.032 | 0.098 | 1.3 |
| | July 2014 | | | 1.9 | 3.4 | | | 1.1 |
| GOB2 | June 2013 | 21°59.5' E, 64°11.6' N | 111 | 1.4 | 4.0 | 0.031 | 0.132 | 1.9 |
| | July 2014 | | | 4.9 | 4.1 | | | 1.3 |
| GOB3 | June 2013 | 18°33.2' E, 62°07.1' N | 91 | 2.6 | 6.0 | 0.022 | 0.124 | 1.6 |
| | July 2014 | | | 3.2 | 6.2 | | | 0.80 |
| RA2 | July 2014 | 22°26.8' E, 65°43.8' N | 12.5 | 8.4 | 2.6 | 0.029 | 0.155 | 0.24 |




**Table 2: Average rates of potential denitrification and anammox from anoxic slurry incubation expressed per gram of wet sediment with associated standard errors (SE, n=15). Average abundances of PC-monoether ladderanes in the top 4 cm of sediment with associated SE (n=4). DW is sediment dry weight and WW is sediment wet weight.**

| Station | Sampling season | Denitrification (nmol N $g^{-1}$ $h^{-1}$) | | Anammox (nmol N $g^{-1}$ $h^{-1}$) | | Ladderanes (pg $g^{-1}$ DW) | | Ladderanes (pg $g^{-1}$ WW) | |
| | | average | SE | average | SE | average | SE | average | SE |
|---|---|---|---|---|---|---|---|---|---|
| GOB1 | June 2013 | | | | | 1073 | 932 | 177 | 149 |
| | July 2014 | 1.038 | 0.060 | 0.244 | 0.023 | 267 | 152 | 51 | 28 |
| GOB2 | June 2013 | | | | | 264 | 148 | 33 | 17 |
| | July 2014 | 1.856 | 0.072 | 0.592 | 0.018 | 34 | 14 | 6 | 2 |
| GOB3 | June 2013 | | | | | 70 | 22 | 18 | 4 |
| | July 2014 | 3.469 | 0.210 | 0.793 | 0.018 | 365 | 128 | 129 | 52 |
| RA2 | July 2014 | 1.499 | 0.031 | 0.286 | 0.003 | 57 | 22 | 7 | 3 |





**Figures**

**Figure 1: Location of the four sampling stations in the Gulf of Bothnia and bathymetry of its two sub-basins, the**

5   **Bothnian Bay and the Bothnian Sea.**







**Figure 2:** Profiles of organic carbon ($C_{org}$) and nitrogen (N) content in the sedimentary solid phase at station GOB1 (a), GOB2 (b) and GOB3 (c) in 2013; and GOB1 (d), GOB2 (e), GOB3 (f), and RA2 (g) in 2014. All values are expressed as $C_{org}$ or N percentage of sediment dry weight (DW).





**Figure 3: Porewater concentration profiles of ammonium, nitrate, nitrite and dissolved organic nitrogen (DON) as a function of depth in the sediment of GOB1 (a), GOB2 (b), and GOB3 (c) in 2013; and GOB1 (d), GOB2 (e), GOB3 (f), and RA2 (g) in 2014. Values on top of each profile represent bottom water samples. Note a second x-axis with**

5    **different scale for (e).**




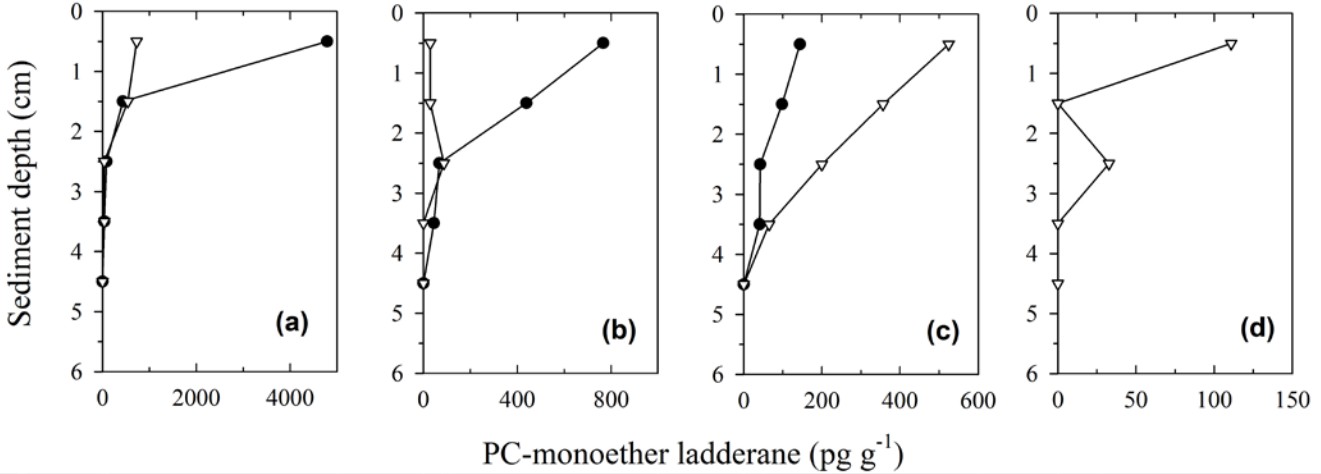

**Figure 4: Abundance profiles of PC-C$_{20}$-[3]-ladderane monoether (pg g$^{-1}$ sediment dry weight) as a function of depth at GOB1 (a); GOB2 (b); GOB3 (c); RA2 (d). Black dots represent values from 2013 while white triangles represent values from 2014. Please note different x-axes.**




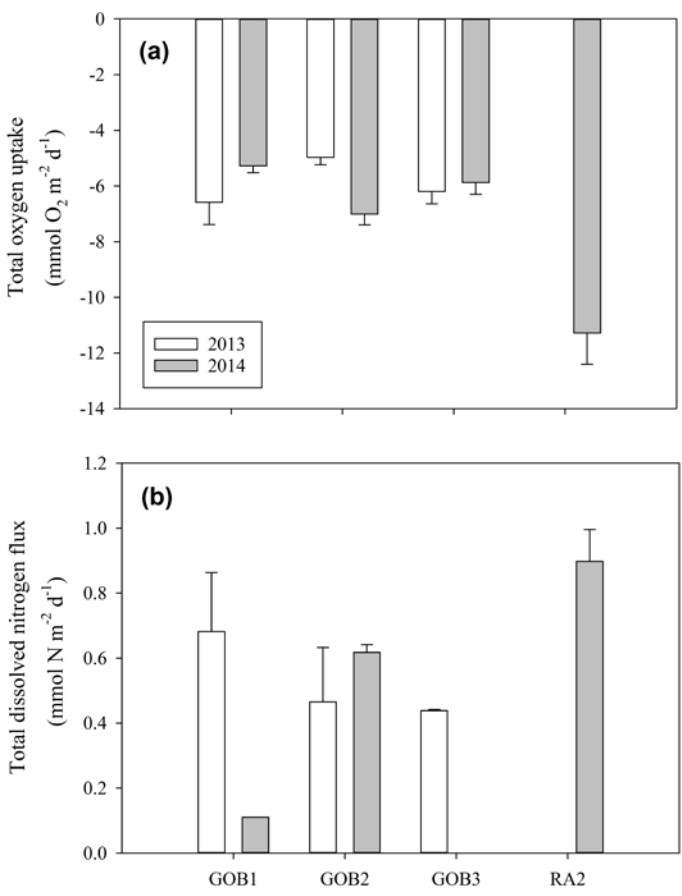

**Figure 5: Total oxygen uptake (TOU) by the sediment (a), and flux of total dissolved nitrogen (TDN) across the sediment-water interface (b) measured at the four stations by in situ incubations with benthic chamber landers. Bars represent average value and error bars represent standard errors. White bars and grey bars represent 2013 and 2014 rates, respectively. At station GOB3, TDN fluxes in 2014 were non-significant.**



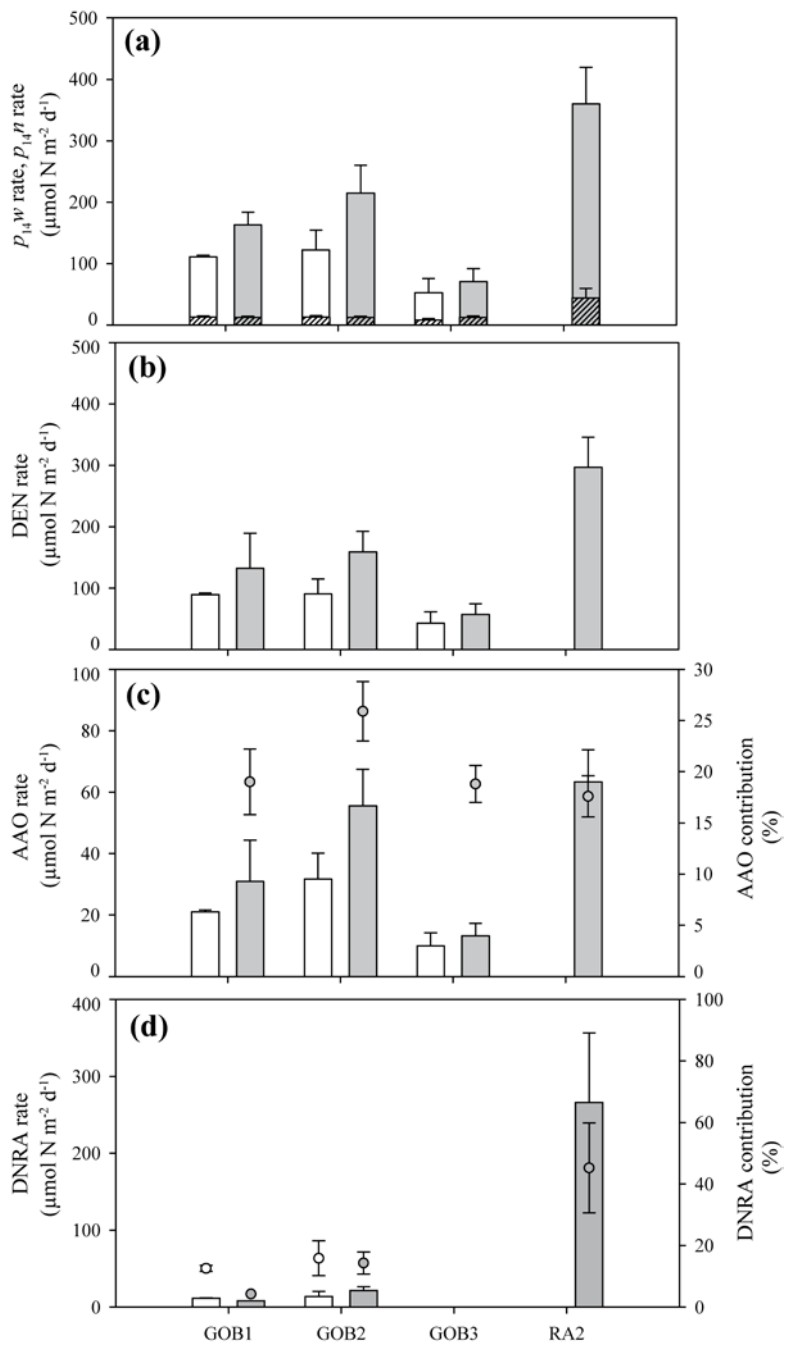

**Figure 6: Average rates of NO$_x^-$-reducing processes measured at the four stations by in situ incubations with benthic chamber landers amended with $^{15}$N-nitrate, where white and gray bars represent 2013 and 2014 rates, respectively, with associated standard errors: (a) Shaded bars represent rates of N$_2$ production depending on the water column**





nitrate ($p_{14}w$) while un-shaded bars represent rates of $N_2$ production coupled to nitrification ($p_{14}n$); (b) Bars represent total denitrification (DEN) rates; (c) Bars represent total anammox (AAO) rates and refer to the left Y-axis while grey dots represent average contribution of anammox to total $N_2$ production ($p_{14}$) and refer to the right Y-axis; (d) Bars represent total DNRA rates and refer to the left Y-axis. White and grey dots represent contribution of DNRA to

5  total nitrate reduction (DEN rate + DNRA rate) and refer to the right Y-axis for year 2013 and 2014, respectively. DNRA rates at GOB3 were not significant.



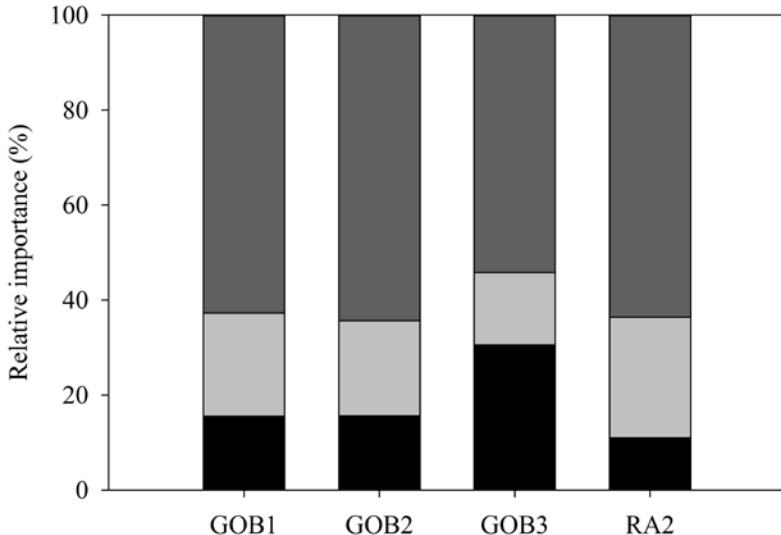

**Figure 7: Relative importance in percentage of sediment burial (black bars), $N_2$ production (light grey bars) and nitrogen recycling (dark grey bars) as N sink and source at the four stations. Processes are expressed on a molar (N) scale. See text for calculations.**

