# Peer review of "The fate of fixed nitrogen in marine sediments with low organic loading: an in situ study"

_Biogeosciences, 2016_

## Referee Comment (RC1) · Anonymous Referee #1 · 12 Oct 2016

General Comments The authors present a study in which they quantified the fate of fixed nitrogen in sediments of a cold, oligotrophic system. The authors used 15N tracers and a combination of in situ incubations using a benthic lander and ex situ sediment core and slurry incubations. The authors are the first to simultaneously measure rates of denitrification, anammox, and DNRA in oligotrophic sediments. They accomplish this using in situ lander incubations, which are logistically difficult to perform, but may actually provide more accurate estimates of in situ rates than traditional core or slurry incubations. The authors found that denitrification dominated N2 production, but anammox bacteria were also active, accounting for 18-26% of N2 production. The authors also measured detectable DNRA and found that DNRA rates were highest, and comparable to denitrification rates, at the shallow coastal station. A sediment nitrogen budget was constructed and indicated that, despite the N2 production measured at the

stations, the primary fate of sediment organic nitrogen in the summer is recycling and efflux as TDN back into the overlying water. Lastly, this study compared concentrations of ladderane lipids, a biomarker for anammox bacteria, to anammox rates and found no correlation between the two. These datasets are sparse in the literature, so this is an informative contribution to the scientific community studying anammox.

Overall, I think the authors addressed important questions related to sediment nitrogen cycling that will be of interest to many readers of this journal. The paper is very well written and organized clearly. I am comfortable with the conclusions and support publication of this manuscript with minor edits, as detailed below.

Specific Comments p.1, line 12 insert "the" before "global"

p.2, line 5 delete "to" before "∼45%"

p.2, line 8 define the abbreviation "DNRA" the first time it's used in the text body

p.2, line 13 insert "the" before "electron"

p.2, lines 23-24 It would be helpful if you mention briefly the link between Mn and anammox, since it is related to your hypotheses and your interpretation of your results.

p.2, line 28 define the abbreviation "GOB" the first time it's used

p.3, line 1 I suggest replacing "happen" with "occur"

p. 3, line 7 suggested change: "….we hypothesize that we will measure low benthic N cycling rates…"

p.3, line 9 change to "porewater," (one word) to be consistent with the rest of the text

p.4, lines 27-28 It would be helpful here if you could define what the average (or range of) water height(s) above the sediment surface was for the lander incubations. No need to list it for every incubation, just give the reader an idea of how much water volume was involved in these incubations.
p.6, line 28 Is the 75uM concentration for the sum of 15NH4+ + 14NO3- or for each of the N species?

p.7, lines 18-20 For clarity, I suggest you present the r-IPT equations from Risgaard-Petersen et al. (2003) so that readers who are unfamiliar with them can understand how you get from p29N2 and p30N2 and ra to p14. This will also give you a chance to define p14 explicitly, and describe how it represents N2 produced without the 15N addition, i.e., actual N2 production. Many unfamiliar with IPT think that the added 15NO3- will stimulate denitrification and that those rates are included in your results, when in actuality the IPT approach allows one to separate p14 (actual) from total N2 production from 15N and 14N (potential).

Eqn. 2 Somewhere here in the text describing eqn. 2 you should state clearly that p14sl includes both water and sediment p14.

p. 7, lines 25-26 I understand why you have to use the same Fwc measured in 2014 for the 2013 calculations—you don't have the sediment core incubations from 2013. I'm just not convinced that the Fwc values would be consistent from 2013 to 2014. Your rates (denitrification, anammox, O2, TDN, etc.) as well as OPD show year-to-year variability, so it would not be surprising to me if the Fwc values were variable. Perhaps here (or elsewhere) you could defend this assumption in a bit more detail and discuss the potential implications for your calculated rates?

p.8, line 1 Since you use the term "ra" here, and it's a widely used term to describe the contribution of anammox to total N2 production, I suggest you use it throughout the rest of the text and tables/figures.

p.8, lines 15-16 I have read this section multiple times, and I still am unsure what this sentence means. I think you're saying that you have to use the Fwc calculated from the p14 values for this NH4+ calculation. If p15NH4+ was not detected in just one of the incubations (GOB1-3), why couldn't you use the p15NH4+ fluxes from all of the other incubations? At least they're still related to the parameter you're working with (NH4+).

How will using the Fwc derived from the p14 values affect the calculated NH4+ rates?

p. 10, line 13 Insert "from" after "ranged"

p. 10, line 24 The sentence "Between the four stations. . ..>GOB3." reads awkwardly. I suggest changing to "Downcore NH4+ concentrations were greatest in RA2, followed by GOB2 >= GOB1>GOB3."

p.11, line 5 Replace "GOB3" with "GOB2"

p. 11, lines 13-15 The sentence "The facts that . . .supported by DON." is awkwardly worded, making it difficult to understand its meaning.

p.11, lines 17-18 reword to ". . .with Fwc values of 0.26, 0.23. . ."

p.11, line 22 At the end of this paragraph, I suggest you present the ra values from the slurry incubations (also include in Table 2), since that's really the main point of doing the slurries. It's fine to keep the data in Figure 6 since it's relevant to the discussion of the other NRPs. But I think the data should be first introduced here to make it clear where that data come from.

p.12, lines 19-22 The sentence "The 15N isotope pairing technique. . .is low.." is awkwardly worded, making it difficult to follow.

p. 13, line 7 Delete "for" before "potential"

p. 13, line 18 Replace "Alike" with "Like"

p.13, line 32 Why did you pool all of the data from 0-4cm for the ladderane concentrations to compare to the rate values? The values are highly variable from surface to 4cm. Did you try just using data from anoxic sediments, where anammox may have been occurring? Or depths where NO2- and NH4+ were present? I wonder if you would have seen a better correlation between the ladderane concentrations and the rates. It would be helpful to include some discussion of this.

p. 14, lines 17-18 You mention here that H2S was never detected in the sediment porewater, but you do not present that data anywhere. I suggest you mention it briefly in the results section since you took the time to describe the microsensor method.

p. 15, line 5 Replace "process" with "proceeds"

p. 15, line 7 Replace "being" with "at"

p. 15, line 13 Delete "eventually"

p. 15, line 16 Replace "upscale" with "scale up"

p. 15, lines 15-17 I'm unsure what conservative method you are referring to. It would be helpful to explain briefly here since it's important enough to bring up in your discussion.

p. 16, line 3 Reword to "The removal rate and the recycling rate were constrained by. . ."

p. 16, line 10 Replace "prove" with "suggest"

p. 16, line 14 Replace "basin-wise" with "basin-wide"

p. 16, lines 17-19 You briefly mention the contribution of DNRA to the TDN flux here, but I think it would be helpful to present the data in Fig. 7 so that the reader can get a feel of interstataion variability.

Figure 2 Make sure to note which symbols are N vs. C (black vs. white).

Figure 6 (c) The y-axis labeled "AAO contribution" should be changed to "ra", as discussed above. Also, the caption for panel (a) should replace "Shaded" with "Hatched" so as not to be confused with the gray shaded bars (2014).

Figure 7 In the caption, replace "nitrogen cycling" with "TDN efflux" since that's more accurate.

---

## Referee Comment (RC2) · Anonymous Referee #2 · 9 Dec 2016

The authors present a high-quality dataset on nitrogen cycling in coastal sediments with a low carbon loading. The manuscript is generally well written and based on a high-quality dataset comprising in situ flux measurements, incubations experiments to partitioning nitrogen flows and some basic background data (ladderane lipids as biomarker for Anammox, burial of nitrogen using 210Pb excess, etc).. The conclusions are largely confirming our existing view of nitrogen biogeochemistry in low carbon coastal sediments and such present a useful addition to the literature. I suggest the authors to articulate their DON flux findings a little more.

Although the writing is generally clear, some fine tuning and precision of wording would improve this very good manuscript further. - insert hyphens for multi-word adjectives: e.g. bottom-water salinity. - one the one and on the other hand always come together - sometimes the logic of sentences needs improvement, e.g. p3, l. 9-10: pore-water

chemistry is the result of N cycling processes; anammox biomarker reflect cycling processes but do not control it, etc.etc. Another example: p. 12, l. 25: our rates therefore represent in situ conditions. Rate reported are representative for the in situ rates. Rates do not represent conditions.

Oligotrophic marine sediments: is that the right term? Water column ecosystems are considered eutrophic or oligotrophic, but sediments are usually classified as low or high carbon loading systems. Nutrient concentrations are quite high in sediment, including the ones reported here. Moreover, can you use the term oligotrophic for sediments with an oxygen penetration depth of less than 2 cm? Not convincing. > 75% of the seafloor has larger OPD.

The authors emphasize somewhat the peculiarities of low temperature conditions, e.g. p. 2, l. 19, but are all deep-sea systems not cold. Consequently there are quite some studies on DNRA in cold systems along ocean margins. Rewrite the text. Moreover, why should temperature matter so much? A permanently cold system will function well, in the end supply of oxidants and reduced substances set the stage.

The material and methods section is very detailed and sometime too much detailed knowledge is expected from the reader: all the abbreviations, etc. Perhaps a few lines on explaining the principle of the approaches would better guide the reader through the details.

On page 8, it is mentioned that C and N were measured before and after HCL treatment. Two remarks: (1) this is the wrong reference because Verardo et al. used sulfurous acid rather than HCl and (2) communicate to the reader that you report only total nitrogen and organic carbon in this manuscript. You made the right choice of not using Norg because of acidification artifacts.

Burial rates are based on sediment burial rates inferred from 210Pb excess measurements. Although you touch upon the issue of bioturbation in the material and methods sections and conclude that you can ignore it, lateron you present visual faune observations suggesting otherwise. Communicate to the reader that burial rates may be inflated because of bioturbation, in particular at stations.. Even better show the 210Pbexcess profiles in the appendix/supplementary info.

Minor corrections: - p. 1, l. 12: on the global - p. 1, l. 13: most scientific investigations have increased the last few years because the scientific community has grown. Reformulate. - P. 1, l. 17: burial rates were not experimentally determined: they were inferred from 210Pbexcess observations - P. 1, l. 24: clarify here that you mean total dissolved fixed nitrogen. - P. 2, l. 26: southern and central Baltic Sea are among the . . . - P. 3, l. 2: but do not report anammox - P. 4, l. 30: control or output? - P. 8, l. 11: an dimensionless linear sorption coefficient - P.10, l. 19: depth-interval weighted average porosities? - P. 12, l. 15: give the most accurate.. - P. 13, l. 17-19: why this role of latitude: is this the cause? I guess that coastal-deep-sea gradient is more important than latitudinal.

---

## Author Comment (AC1) · 12 Dec 2016

Here we present our answers (marked AC) below the original referees' comments (RC).

Anonymous Referee #1

RC - General Comments The authors present a study in which they quantified the fate of fixed nitrogen in sediments of a cold, oligotrophic system. The authors used 15N tracers and a combination of in situ incubations using a benthic lander and ex situ sediment core and slurry incubations. The authors are the first to simultaneously measure rates of denitrification, anammox, and DNRA in oligotrophic sediments. They accomplish this using in situ lander incubations, which are logistically difficult to perform, but may actually provide more accurate estimates of in situ rates than traditional core or slurry incubations. The authors found that denitrification dominated N2 production, but

anammox bacteria were also active, accounting for 18-26% of N2 production. The authors also measured detectable DNRA and found that DNRA rates were highest, and comparable to denitrification rates, at the shallow coastal station. A sediment nitrogen budget was constructed and indicated that, despite the N2 production measured at the stations, the primary fate of sediment organic nitrogen in the summer is recycling and efflux as TDN back into the overlying water. Lastly, this study compared concentrations of ladderane lipids, a biomarker for anammox bacteria, to anammox rates and found no correlation between the two. These datasets are sparse in the literature, so this is an informative contribution to the scientific community studying anammox. Overall, I think the authors addressed important questions related to sediment nitrogen cycling that will be of interest to many readers of this journal. The paper is very well written and organized clearly. I am comfortable with the conclusions and support publication of this manuscript with minor edits, as detailed below.

AC - Thank you for the very detailed and insightful analysis of our manuscript and for the positive reception of our work. We really appreciate the efforts and the time the referee invested in improving the manuscript.

RC - Specific Comments p.1, line 12 insert "the" before "global" p.2, line 5 delete "to" before "âĹij45%" p.2, line 8 define the abbreviation "DNRA" the first time it's used in the text body p.2, line 13 insert "the" before "electron" p.2, lines 23-24 It would be helpful if you mention briefly the link between Mn and anammox, since it is related to your hypotheses and your interpretation of your results. p.2, line 28 define the abbreviation "GOB" the first time it's used p.3, line 1 I suggest replacing "happen" with "occur"

AC – We will be glad to consider these specific comments in the revised manuscript.

RC - p. 3, line 7 suggested change: ". . ..we hypothesize that we will measure low benthic N cycling rates. . ."

AC – In our opinion it does not sound correct to use the future tense in this context. We therefore propose the following compromise, at the infinitive form: ". . .we hypothesize

to measure...". We will then change all the verbs of this last paragraph to the infinitive form or present tense for consistency.

RC - p.3, line 9 change to "porewater," (one word) to be consistent with the rest of the text p.4, lines 27-28 It would be helpful here if you could define what the average (or range of) water height(s) above the sediment surface was for the lander incubations. No need to list it for every incubation, just give the reader an idea of how much water volume was involved in these incubations. p.6, line 28 Is the 75uM concentration for the sum of 15NH4+ + 14NO3- or for each of the N species?

AC – We will address these minor edits in the revised manuscript.

RC - p.7, lines 18-20 For clarity, I suggest you present the r-IPT equations from Risgaard-Petersen et al. (2003) so that readers who are unfamiliar with them can understand how you get from p29N2 and p30N2 and ra to p14. This will also give you a chance to define p14 explicitly, and describe how it represents N2 produced without the 15N addition, i.e., actual N2 production. Many unfamiliar with IPT think that the added 15NO3- will stimulate denitrification and that those rates are included in your results, when in actuality the IPT approach allows one to separate p14 (actual) from total N2 production from 15N and 14N (potential).

AC - The r-IPT equations from Risgaard-Petersen et al. (2003) will be added to the text in order to explain how, for example, p14lan was calculated. For conciseness, however, we won't present the calculations also for p14wc and p14sl as the reader will have sufficient information to understand that they were calculated in the same fashion as for p14lan but from sediment core incubations (water phase and slurry phase, respectively).

RC - Eqn. 2 Somewhere here in the text describing eqn. 2 you should state clearly that p14sl includes both water and sediment p14.

AC - We will add in the text above Eq. 2 that slurried phase means water plus sediment.

RC - p. 7, lines 25-26 I understand why you have to use the same Fwc measured in 2014 for the 2013 calculationsâA ËŸTyou don't have the sediment core incubations from 2013. ËĞI'm just not convinced that the Fwc values would be consistent from 2013 to 2014. Your rates (denitrification, anammox, O2, TDN, etc.) as well as OPD show year-to-year variability, so it would not be surprising to me if the Fwc values were variable. Perhaps here (or elsewhere) you could defend this assumption in a bit more detail and discuss the potential implications for your calculated rates?ÂÍ

AC - We acknowledge the referee for this adequate comment. We cannot indeed exclude that Fwc in 2013 could have been slightly lower than those we measured in 2014 because the oxygen penetration depths were higher in 2013 than in 2014 (Table 1), suggesting that N2 production happened deeper in the sediment. A low underestimation of in situ N2 production in 2013 cannot be excluded. We will add this argument in the Results when we present our measured Fwc values.

RC - p.8, line 1 Since you use the term "ra" here, and it's a widely used term to describe the contribution of anammox to total N2 production, I suggest you use it throughout the rest of the text and tables/figures.

AC – We will make this edit throughout the text and in Fig. 6.

RC - p.8, lines 15-16 I have read this section multiple times, and I still am unsure what this sentence means. I think you're saying that you have to use the Fwc calculated from the p14 values for this NH4+ calculation. If p15NH4+ was not detected in just one of the incubations (GOB1-3), why couldn't you use the p15NH4+ fluxes from all of the other incubations? At least they're still related to the parameter you're working with (NH4+). How will using the Fwc derived from the p14 values affect the calculated NH4+ rates?

AC - As we stated it in the Results, DNRA could not be determined in sediment core incubations because p15NH4+ was not detectable in our time courses at three (GOB1, GOB2, GOB3) out of four stations. In sediment cores incubations, p15NH4+ could

only be detected at station RA2. Using the Fwc from p15NH4+ at RA2 for all the other stations would result in much higher, and probably unrealistic, DNRA rates than those we estimated now by using Fwc from whole core's p14. However, we agree with the referee that using Fwc from p14 may not be 100% representative of the in situ, actual rate. Thus, in the Discussion, we will further highlight the fact that our up-scaled DNRA rate may represent an underestimation of the actual DNRA rates.

RC - p. 10, line 13 Insert "from" after "ranged" p. 10, line 24 The sentence "Between the four stations. . ..>GOB3." reads awkwardly. I suggest changing to "Downcore NH4+ concentrations were greatest in RA2, followed by GOB2 >= GOB1>GOB3." p.11, line 5 Replace "GOB3" with "GOB2" p. 11, lines 13-15 The sentence "The facts that . . .supported by DON." is awkwardly worded, making it difficult to understand its meaning. p.11, lines 17-18 reword to ". . .with Fwc values of 0.26, 0.23. . ." p.11, line 22 At the end of this paragraph, I suggest you present the ra values from the slurry incubations (also include in Table 2), since that's really the main point of doing the slurries. It's fine to keep the data in Figure 6 since it's relevant to the discussion of the other NRPs. But I think the data should be first introduced here to make it clear where that data come from. p.12, lines 19-22 The sentence "The 15N isotope pairing technique. . .is low." is awkwardly worded, making it difficult to follow. p. 13, line 7 Delete "for" before "potential" p. 13, line 18 Replace "Alike" with "Like"

AC – We will gladly address these minor edits in the revised manuscript.

RC - p.13, line 32 Why did you pool all of the data from 0-4cm for the ladderane concentrations to compare to the rate values? The values are highly variable from surface to 4cm. Did you try just using data from anoxic sediments, where anammox may have been occurring? Or depths where NO2- and NH4+ were present? I wonder if you would have seen a better correlation between the ladderane concentrations and the rates. It would be helpful to include some discussion of this.

AC - We pooled the ladderane data from 0-4 cm because this is the approach that is

often used in literature. However, we agree with the referee that this is not the best approach when values are highly variable, as in our situation. We have now correlated the potential anammox rates with (1) the average ladderane abundances and (2) with the ladderane abundances in the anoxic sediments layer (1.5-3.5 cm), which coincides with the layer sampled for anoxic slurry experiments for anammox potential. Yet, we cannot see any significant correlations. We will add this part to the Results. In the Discussion we will explain that these non-significant correlations might be due to the low number of observations.

RC - p. 14, lines 17-18 You mention here that H2S was never detected in the sediment porewater, but you do not present that data anywhere. I suggest you mention it briefly in the results section since you took the time to describe the microsensor method. p. 15, line 5 Replace "process" with "proceeds" p. 15, line 7 Replace "being" with "at" p. 15, line 13 Delete "eventually" p. 15, line 16 Replace "upscale" with "scale up" p. 15, lines 15-17 I'm unsure what conservative method you are referring to. It would be helpful to explain briefly here since it's important enough to bring up in your discussion. p. 16, line 3 Reword to "The removal rate and the recycling rate were constrained by. . ." p. 16, line 10 Replace "prove" with "suggest" p. 16, line 14 Replace "basin-wise" with "basin-wide" p. 16, lines 17-19 You briefly mention the contribution of DNRA to the TDN flux here, but I think it would be helpful to present the data in Fig. 7 so that the reader can get a feel of interstataion variability. Figure 2 Make sure to note which symbols are N vs. C (black vs. white). Figure 6 (c) The y-axis labeled "AAO contribution" should be changed to "ra", as discussed above. Also, the caption for panel (a) should replace "Shaded" with "Hatched" so as not to be confused with the gray shaded bars (2014). Figure 7 In the caption, replace "nitrogen cycling" with "TDN efflux" since that's more accurate.

AC – We will consider these specific comments in the revised manuscript. Your input was very much appreciated.

---

## Author Comment (AC2) · 12 Dec 2016

Here we present our answers (marked AC) below the original referees' comments (RC).

Anonymous Referee #2

RC - The authors present a high-quality dataset on nitrogen cycling in coastal sediments with a low carbon loading. The manuscript is generally well written and based on a high-quality dataset comprising in situ flux measurements, incubations experiments to partitioning nitrogen flows and some basic background data (ladderane lipids as biomarker for Anammox, burial of nitrogen using 210Pb excess, etc).. The conclusions are largely confirming our existing view of nitrogen biogeochemistry in low carbon coastal sediments and such present a useful addition to the literature. I suggest the authors to articulate their DON flux findings a little more.

AC – We appreciate the reviewer's acknowledgement of the merits of our work, and we thank them for their insightful and useful comments. However, we do not fully agree that our conclusions are largely confirming existing data, as our results contrast with previous studies suggesting that DNRA was negligible in cold and well-oxygenated sediments with low organic carbon loads. We also believe that the results concerning the DON data are intriguing and novel. We will put more emphasis on the high contribution of the DON flux to the total efflux of fixed nitrogen. We will also discuss some implications of this aspect. High export of DON to the water column may be a reason for the high activity of bacterioplankton and the dominance of heterotrophy vs. autotrophy found in the waters of the Gulf of Bothnia (Algesten et al. 2004 - Global Biogeochem. Cycles).

RC - Although the writing is generally clear, some fine tuning and precision of wording would improve this very good manuscript further. - insert hyphens for multi-word adjectives: e.g. bottom-water salinity. - one the one and on the other hand always come together - sometimes the logic of sentences needs improvement, e.g. p3, l. 9-10: pore-water chemistry is the result of N cycling processes; anammox biomarker reflect cycling processes but do not control it, etc.etc. Another example: p. 12, l. 25: our rates therefore represent in situ conditions. Rate reported are representative for the in situ rates. Rates do not represent conditions.

AC – We appreciate these corrections and we will reword the text accordingly.

RC - Oligotrophic marine sediments: is that the right term? Water column ecosystems are considered eutrophic or oligotrophic, but sediments are usually classified as low or high carbon loading systems. Nutrient concentrations are quite high in sediment, including the ones reported here. Moreover, can you use the term oligotrophic for sediments with an oxygen penetration depth of less than 2 cm? Not convincing. > 75% of the seafloor has larger OPD.

AC – We agree with the reviewer that "oligotrophic" is not the most suitable adjective

to describe marine sediments, although it is commonly used in literature. We suggest the following alternative title: "The fate of fixed nitrogen in marine sediments with low carbon loads: an in situ study".

RC - The authors emphasize somewhat the peculiarities of low temperature conditions, e.g. p. 2, l. 19, but are all deep-sea systems not cold. Consequently there are quite some studies on DNRA in cold systems along ocean margins. Rewrite the text. Moreover, why should temperature matter so much? A permanently cold system will function well, in the end supply of oxidants and reduced substances set the stage.

AC – Temperate coastal sediments, except for those of the high Arctic/Antarctic, have seasonal temperature variations that may affect biogeochemical processes. In other cold Baltic Sea sediments, for example, temperature was shown to significantly affect nitrogen cycling processes and the partitioning between denitrification and DNRA rates (Bonaglia et al. 2014 – Biogeochemistry). Moreover, DNRA bacteria isolated in Arctic fjord sediments had their highest optimal growth rate at 18 °C, while denitrifiers had their optima at 0 °C (Canion et al. 2013 - Environ. Microbiol.). Even in the permanently cold (< 10 °C) GOB sediments we have temperature fluctuations, distinguishing them from the Artic and deep-sea sediments. To date, we are not aware of any single study reporting on significant DNRA activity in year-round cold sediments, either from coastal setups or the open sea. This is further corroborated by the study just published by McTigue et al. (2016 – Nature Comm.), which showed that denitrification was one to two orders of magnitude greater than DNRA in Alaskan Arctic shelf sediments. Thus, one of the main messages of our paper is that significant DNRA activity cannot be excluded a priori in cold, oligotrophic systems.

RC - The material and methods section is very detailed and sometime too much detailed knowledge is expected from the reader: all the abbreviations, etc. Perhaps a few lines on explaining the principle of the approaches would better guide the reader through the details.

AC – We believe that it is preferable to describe Methods in details rather than omitting important steps of the operations in this type of scientific works with novel and complex experimental setups. However, in the revised manuscript, we will shorten the 210Pb and ladderane parts, which are already been described in details by others before. We will also introduce the main principles behind each of these methodologies.

RC - On page 8, it is mentioned that C and N were measured before and after HCL treatment. Two remarks: (1) this is the wrong reference because Verardo et al. used sulfurous acid rather than HCl and (2) communicate to the reader that you report only total nitrogen and organic carbon in this manuscript. You made the right choice of not using Norg because of acidification artifacts.

AC – The procedure by Verardo et al. was referenced because of the type of detector used (flash combustor by a Carlo Erba elemental analyzer). We will specify that we slightly modified the sample preparation method and that only the Corg and N data are presented in the paper.

RC - Burial rates are based on sediment burial rates inferred from 210Pb excess measurements. Although you touch upon the issue of bioturbation in the material and methods sections and conclude that you can ignore it, lateron you present visual faune observations suggesting otherwise. Communicate to the reader that burial rates may be inflated because of bioturbation, in particular at stations.. Even better show the 210Pbexcess profiles in the appendix/supplementary info.

AC – We exclude that in this type of sediments bioturbation may have biased burial rates. The macrofaunal organisms retrieved in the benthic chambers and in the sediment cores were almost exclusively specimens of Monoporeia affinis, a small amphipod that was found either swimming in the water column or colonizing the upper 3-4 cm of the sediment. The abundances of the deep burrower Marenzelleria spp. were negligible and their effect on the 210Pb distribution was therefore minimal. Moreover, macrofauna was completely absent at RA2 and at the GOB stations sediments were

laminated below 5-6 cm depth, which clearly exclude particle mixing below that depth.

RC - Minor corrections: - p. 1, l. 12: on the global - p. 1, l. 13: most scientific investigations have increased the last few years because the scientific community has grown. Reformulate. - P. 1, l. 17: burial rates were not experimentally determined: they were inferred from 210Pbexcess observations - P. 1, l. 24: clarify here that you mean total dissolved fixed nitrogen. - P. 2, l. 26: southern and central Baltic Sea are among the : : : - P. 3, l. 2: but do not report anammox - P. 4, l. 30: control or output? - P. 8, l. 11: an dimensionless linear sorption coefficient - P.10, l. 19: depth-interval weighted average porosities? - P. 12, l. 15: give the most accurate.. - P. 13, l. 17-19: why this role of latitude: is this the cause? I guess that coastal-deep-sea gradient is more important than latitudinal.

AC – We will be glad to consider these specific comments in the revised manuscript.

---

## Author Response (AR1)

**This file contains:**

- a point-by-point response to the referees, which includes a description of the relevant changes;

- a marked-up manuscript version, i.e., the manuscript with Track Changes activated.

**Dear editor,**

**We are pleased to present our revised version of the manuscript "The fate of fixed nitrogen in marine sediments with low organic loading: an in situ study". This version incorporates the suggestions provided by the two anonymous reviewers, whom we would like to thank for their detailed analysis of our work. We think that, with their suggestions and criticism, this revised version of the paper has substantially improved. We have addressed the reviewers' comments point-by-point and we presented our answers (in blue) below their original comments (in black).**

Anonymous Referee #1

General Comments The authors present a study in which they quantified the fate of fixed nitrogen in sediments of a cold, oligotrophic system. The authors used 15N tracers and a combination of in situ incubations using a benthic lander and ex situ sediment core and slurry incubations. The authors are the first to simultaneously measure rates of denitrification, anammox, and DNRA in oligotrophic sediments. They accomplish this using in situ lander incubations, which are logistically difficult to perform, but may actually provide more accurate estimates of in situ rates than traditional core or slurry incubations. The authors found that denitrification dominated N2 production, but anammox bacteria were also active, accounting for 18-26% of N2 production. The authors also measured detectable DNRA and found that DNRA rates were highest, and comparable to denitrification rates, at the shallow coastal station. A sediment nitrogen budget was constructed and indicated that, despite the N2 production measured at the stations, the primary fate of sediment organic nitrogen in the summer is recycling and efflux as TDN back into the overlying water. Lastly, this study compared concentrations of ladderane lipids, a biomarker for anammox bacteria, to anammox rates and found no correlation between the two. These datasets are sparse in the literature, so this is an informative contribution to the scientific community studying anammox.

Overall, I think the authors addressed important questions related to sediment nitrogen cycling that will be of interest to many readers of this journal. The paper is very well written and organized clearly. I am comfortable with the conclusions and support publication of this manuscript with minor edits, as detailed below.

**We would like to thank the reviewer for their detailed and insightful analysis of our work. We really appreciate their efforts in improving the manuscript, and we are happy to read their acknowledgment of the manuscript merits.**

Specific Comments p.1, line 12 insert "the" before "global"
**Edit made.**

p.2, line 5 delete "to" before "          ∼ 45%"
**Edit made.**

p.2, line 8 define the abbreviation "DNRA" the first time it's used in the text body
**We have explained the acronym "DNRA", as well as "N", "GOB" and "TDN" at their first use in the text body, although these acronyms were already defined in the Abstract.**

p.2, line 13 insert "the" before "electron"
**Edit made.**

p.2, lines 23-24 It would be helpful if you mention briefly the link between Mn and anammox, since it is related to your hypotheses and your interpretation of your results.

**We have presented the possible link between high Mn concentrations and high anammox contribution/low denitrification contribution.**

p.2, line 28 define the abbreviation "GOB" the first time it's used
**Abbreviation explained.**

p.3, line 1 I suggest replacing "happen" with "occur"
**Edit made.**

p. 3, line 7 suggested change: ". . ..we hypothesize that we will measure low benthic N cycling rates. . ."
**In our opinion it does not sound correct to use the future tense in this context. The past tense in English is often used to talk about hypotheses. For consistency, we have left the verb at the past tense here.**

p.3, line 9 change to "porewater," (one word) to be consistent with the rest of the text
**Edit made.**

p.4, lines 27-28 It would be helpful here if you could define what the average (or range of) water height(s) above the sediment surface was for the lander incubations. No need to list it for every incubation, just give the reader an idea of how much water volume was involved in these incubations.
**We have now included the range of the incubated water volumes.**

p.6, line 28 Is the 75uM concentration for the sum of 15NH4+ + 14NO3- or for each of the N species?
**We have now specified that it refers to each of the N species.**

p.7, lines 18-20 For clarity, I suggest you present the r-IPT equations from Risgaard-Petersen et al. (2003) so that readers who are unfamiliar with them can understand how you get from p29N2 and p30N2 and ra to p14. This will also give you a chance to define p14 explicitly, and describe how it represents N2 produced without the 15N addition, i.e., actual N2 production. Many unfamiliar with IPT think that the added 15NO3- will stimulate denitrification and that those rates are included in your results, when in actuality the IPT approach allows one to separate p14 (actual) from total N2 production from 15N and 14N (potential).
**The r-IPT equations from Risgaard-Petersen et al. (2003) have been added to the text in order to explain how, for example, $p_{14}lan$ was calculated. For conciseness, however, we have decided not to present the calculations also for $p_{14}wc$ and $p_{14}sl$ as the reader has now sufficient information to understand that they were calculated in the same fashion as for $p_{14}lan$ but from sediment core incubations (water phase and slurry phase, respectively).**

Eqn. 2 Somewhere here in the text describing eqn. 2 you should state clearly that p14sl includes both water and sediment p14.
**We have added in the text above Eq. 2 that slurried phase means water plus sediment.**

p. 7, lines 25-26 I understand why you have to use the same Fwc measured in 2014 for the 2013 calculationsâA˘Tyou don't have the sediment core incubations from 2013. ˇI'm just not convinced that the Fwc values would be consistent from 2013 to 2014. Your rates (denitrification, anammox, O2, TDN, etc.) as well as OPD show year-to-year variability, so it would not be surprising to me if the Fwc values were variable. Perhaps here (or elsewhere) you could defend this assumption in a bit more detail and discuss the potential implications for your calculated rates?¨
**We acknowledge the referee for raising an important point here. We cannot indeed exclude that $F_{wc}$ in 2013 could have been slightly lower than those we measured in 2014 because the oxygen penetration depths were**

**higher in 2013 than in 2014 (Table 1), suggesting that $N_2$ production happened deeper in the sediment. A small underestimation of in situ $N_2$ production in 2013 cannot be excluded. We have added this argument in the Results when we present our measured $F_{wc}$ values.**

p.8, line 1 Since you use the term "ra" here, and it's a widely used term to describe the contribution of anammox to total N2 production, I suggest you use it throughout the rest of the text and tables/figures.
**Edit made throughout the text and in Fig. 6.**

p.8, lines 15-16 I have read this section multiple times, and I still am unsure what this sentence means. I think you're saying that you have to use the Fwc calculated from the p14 values for this NH4+ calculation. If p15NH4+ was not detected in just one of the incubations (GOB1-3), why couldn't you use the p15NH4+ fluxes from all of the other incubations? At least they're still related to the parameter you're working with (NH4+). How will using the Fwc derived from the p14 values affect the calculated NH4+ rates?
**As stated in the Results, DNRA could not be determined in sediment core incubations because $p^{15}NH_4^+$ was not detectable in our time courses at three (GOB1, GOB2, GOB3) out of four stations. In sediment core incubations, $p^{15}NH_4^+$ could only be detected at station RA2. Using the Fwc from $p^{15}NH_4^+$ at RA2 for all the other stations would result in much higher, and probably unrealistic, DNRA rates than those we estimated now by using Fwc from whole core's $p_{14}$. However, we agree with the referee that using Fwc from $p_{14}$ may not be 100% representative of the in situ, actual rate. Thus, in the Discussion, we have further highlighted the fact that our up-scaled DNRA rate may represent an underestimation of the actual DNRA rates.**

p. 10, line 13 Insert "from" after "ranged"
**Edit made.**

p. 10, line 24 The sentence "Between the four stations. . ..>GOB3." reads awkwardly. I suggest changing to "Downcore NH4+ concentrations were greatest in RA2, followed by GOB2 >= GOB1>GOB3."
**Edit made.**

p.11, line 5 Replace "GOB3" with "GOB2"
**Thank you for this important correction. We have now replaced GOB3 with GOB2.**

p. 11, lines 13-15 The sentence "The facts that . . .supported by DON." is awkwardly worded, making it difficult to understand its meaning.
**We agree with the referee and have now rephrased this all sentence.**

p.11, lines 17-18 reword to ". . .with Fwc values of 0.26, 0.23. . ."
**Edit made.**

p.11, line 22 At the end of this paragraph, I suggest you present the ra values from the slurry incubations (also include in Table 2), since that's really the main point of doing the slurries. It's fine to keep the data in Figure 6 since it's relevant to the discussion of the other NRPs. But I think the data should be first introduced here to make it clear where that data come from.
**We have now added the ra values at the end of the paragraph and in Table 2.**

p.12, lines 19-22 The sentence "The 15N isotope pairing technique. . .is low.." is awkwardly worded, making it difficult to follow.
**We have now split this sentence into two shorter sentences.**

p. 13, line 7 Delete "for" before "potential"
**Edit made.**

p. 13, line 18 Replace "Alike" with "Like"
**Edit made.**

p.13, line 32 Why did you pool all of the data from 0-4cm for the ladderane concentrations to compare to the rate values? The values are highly variable from surface to 4cm. Did you try just using data from anoxic sediments, where anammox may have been occurring? Or depths where NO2- and NH4+ were present? I wonder if you would have seen a better correlation between the ladderane concentrations and the rates. It would be helpful to include some discussion of this.
**In the previous version of the manuscript we pooled the ladderane data from 0-4 cm because this is the approach that is often used in literature. However, we agree with the referee that this is not the best approach when values are highly variable, as in our situation. We have now correlated the potential anammox rates with (1) the average ladderane abundances and (2) with the ladderane abundances in the anoxic sediments layer (1.5-3.5 cm), which coincides with the layer sampled for anoxic slurry experiments for anammox potential. We have also performed correlations between ladderane abundances and environmental parameters (temperature, water depth, salinity, etc.). Yet, we did not see any statistically significant correlations. We have now added this information to the Results. In the Discussion we have now explained that these non-significant correlations might be due to the low number of observations or to differences in microbial spatial heterogeneity between the sediment used for the lipid analysis and the sediment used for the 15N incubations.**

p. 14, lines 17-18 You mention here that H2S was never detected in the sediment porewater, but you do not present that data anywhere. I suggest you mention it briefly in the results section since you took the time to describe the microsensor method.
**We have added a sentence in the Results to explain that $H_2S$ concentrations were below detection limits.**

p. 15, line 5 Replace "process" with "proceeds"
**Edit made.**

p. 15, line 7 Replace "being" with "at"
**Edit made.**

p. 15, line 13 Delete "eventually"
**Edit made.**

p. 15, line 16 Replace "upscale" with "scale up"
**Edit made.**

p. 15, lines 15-17 I'm unsure what conservative method you are referring to. It would be helpful to explain briefly here since it's important enough to bring up in your discussion.
**We have now explained what source of error we refer to.**

p. 16, line 3 Reword to "The removal rate and the recycling rate were constrained by. . ."
**Edit made.**

p. 16, line 10 Replace "prove" with "suggest"

**Edit made.**

p. 16, line 14 Replace "basin-wise" with "basin-wide"
**Edit made.**

p. 16, lines 17-19 You briefly mention the contribution of DNRA to the TDN flux here, but I think it would be helpful to present the data in Fig. 7 so that the reader can get a feel of interstataion variability.
**We have added the ranges of the relative contribution of DNRA to the TDN fluxes in Figure 7.**

Figure 2 Make sure to note which symbols are N vs. C (black vs. white).
**Edit made.**

Figure 6 (c) The y-axis labeled "AAO contribution" should be changed to "ra", as discussed above. Also, the caption for panel (a) should replace "Shaded" with "Hatched" so as not to be confused with the gray shaded bars (2014).
**Edits made. Thank you.**

Figure 7 In the caption, replace "nitrogen cycling" with "TDN efflux" since that's more accurate.
**Edit made.**

………………………………………………………………………………………………………………………………………………………………………………………………………………………………………………

Anonymous Referee #2

The authors present a high-quality dataset on nitrogen cycling in coastal sediments with a low carbon loading. The manuscript is generally well written and based on a high-quality dataset comprising in situ flux measurements, incubations experiments to partitioning nitrogen flows and some basic background data (ladderane lipids as biomarker for Anammox, burial of nitrogen using 210Pb excess, etc).. The conclusions are largely confirming our existing view of nitrogen biogeochemistry in low carbon coastal sediments and such present a useful addition to the literature. I suggest the authors to articulate their DON flux findings a little more.
**We appreciate the reviewer's acknowledgement of the merits of our work, and we thank her/him for her/his insightful and useful comments. We also believe that the results concerning the DON data are intriguing and novel. We have put more emphasis on the high contribution of the DON flux to the total efflux of fixed nitrogen. This high DON export from the sediments to the water column may be a reason for the high activity of bacterioplankton and the dominance of heterotrophy vs. autotrophy found in the waters of the Gulf of Bothnia (Algesten et al. 2004). We discussed these aspects on page 16 and 17; section 4.3.**

Although the writing is generally clear, some fine tuning and precision of wording would improve this very good manuscript further. - insert hyphens for multi-word adjectives: e.g. bottom-water salinity.
**Edits made.**

- one the one and on the other hand always come together
**Edits made.**

- sometimes the logic of sentences needs improvement, e.g. p3, l. 9-10: pore-water chemistry is the result of N cycling processes; anammox biomarker reflect cycling processes but do not control it, etc.etc. Another example: p. 12, l. 25: our rates therefore represent in situ conditions. Rate reported are representative for the in situ rates. Rates do not represent conditions.

**We appreciated these corrections. We have reworded the text accordingly.**

Oligotrophic marine sediments: is that the right term? Water column ecosystems are considered eutrophic or oligotrophic, but sediments are usually classified as low or high carbon loading systems. Nutrient concentrations are quite high in sediment, including the ones reported here. Moreover, can you use the term oligotrophic for sediments with an oxygen penetration depth of less than 2 cm? Not convincing. > 75% of the seafloor has larger OPD.

**We agree with the reviewer that "oligotrophic" is not the most suitable adjective to describe marine sediments, although it is commonly used in literature. We have changed the title to: "The fate of fixed nitrogen in marine sediments with low organic loading: an in situ study". For consistency, we have also changed the term "oligotrophic sediments" into "low-organic sediments" throughout the text.**

The authors emphasize somewhat the peculiarities of low temperature conditions, e.g. p. 2, l. 19, but are all deep-sea systems not cold. Consequently there are quite some studies on DNRA in cold systems along ocean margins. Rewrite the text. Moreover, why should temperature matter so much? A permanently cold system will function well, in the end supply of oxidants and reduced substances set the stage.

**Temperate coastal sediments, except for those of the high Arctic/Antarctic, have seasonal temperature variations that may affect biogeochemical processes. In other cold Baltic Sea sediments, for example, temperature was shown to significantly affect nitrogen cycling processes and the partitioning between denitrification and DNRA rates (Bonaglia et al. 2014). Moreover, DNRA bacteria isolated in Arctic fjord sediments had their highest optimal growth rate at 18 °C, while denitrifiers had their optima at 0 °C (Canion et al. 2013).**
**Even in the permanently cold (< 10 °C) GOB sediments we have temperature fluctuations, distinguishing them from the Arctic and deep-sea sediments. To date, we are not aware of any single study reporting on significant DNRA activity in year-round cold sediments, either from coastal environments or the open sea. This is further corroborated by the study just published by McTigue et al. (2016), which showed that denitrification was one to two orders of magnitude greater than DNRA in Alaskan Arctic shelf sediments. Thus, one of the main messages of our paper is that significant DNRA activity cannot be excluded *a priori* in cold, oligotrophic systems.**

The material and methods section is very detailed and sometime too much detailed knowledge is expected from the reader: all the abbreviations, etc. Perhaps a few lines on explaining the principle of the approaches would better guide the reader through the details.

**We believe that it is preferable to describe Methods in details rather than omitting important steps of the operations in this type of scientific works with novel and complex experimental setups. However, in the revised manuscript, we have shortened the [210]Pb and ladderane parts, whose protocols are already been described in details by others. We also briefly introduced the main principles behind each of these methodologies.**

On page 8, it is mentioned that C and N were measured before and after HCL treatment. Two remarks: (1) this is the wrong reference because Verardo et al. used sulfurous acid rather than HCl and (2) communicate to the reader that you report only total nitrogen and organic carbon in this manuscript. You made the right choice of not using Norg because of acidification artifacts.

**The procedure by Verardo et al. was referenced because of the type of detector used (flash combustor by a Carlo Erba elemental analyzer). We have specified that we slightly modified the sample preparation method and that only the $C_{org}$ and N data are presented in the paper. We have also motivated why we excluded the $N_{org}$ data presentation. Thanks for this feedback.**

Burial rates are based on sediment burial rates inferred from 210Pb excess measurements. Although you touch upon the issue of bioturbation in the material and methods sections and conclude that you can ignore it, lateron you present visual faune observations suggesting otherwise. Communicate to the reader that burial rates may be inflated because of bioturbation, in particular at stations.. Even better show the 210Pbexcess profiles in the appendix/supplementary info.

**We exclude that in this type of sediments bioturbation may have biased burial rates. The macrofaunal organisms retrieved in the benthic chambers and in the sediment cores were almost exclusively specimens of *Monoporeia affinis*, a small amphipod that was found either swimming in the water column or colonizing the upper 3-4 cm of the sediment. The abundances of the deep burrower *Marenzelleria* spp. were negligible and their effect on the $^{210}$Pb distribution was therefore minimal. Moreover, macrofauna was completely absent at RA2 and at the GOB stations sediments were laminated below 5-6 cm depth, which clearly exclude particle mixing below that depth. We have specified these points both in the Methods (Page 9, Line 9) and in the Results (Page 10, Lines 10-17).**

- Minor corrections: - p. 1, l. 12: on the global
**Edit made.**

- p. 1, l. 13: most scientific investigations have increased the last few years because the scientific community has grown. Reformulate.
**We have edited the first two sentences of the Abstract, which reads much better now.**

- P. 1, l. 17: burial rates were not experimentally determined: they were inferred from 210Pbexcess observations
**Edit made.**

- P. 1, l. 24: clarify here that you mean total dissolved fixed nitrogen.
**Edit made.**

- P. 2, l. 26: southern and central Baltic Sea are among the *: : :*
**Edit made.**

- P. 3, l. 2: but do not report anammox
**Edit made.**

- P. 4, l. 30: control or output?
**The correct word here is output.**

- P. 8, l. 11: an dimensionless linear sorption coefficient
**Edit made.**

- P.10, l. 19: depth-interval weighted average porosities?
**Edit made.**

- P. 12, l. 15: give the most accurate..
**Edit made.**

- P. 13, l. 17-19: why this role of latitude: is this the cause? I guess that coastal-deep-sea gradient is more important than latitudinal.
**We agree with the reviewer that in this context lower latitude is not really the cause of lower anammox rates. We have thus removed this comparison. Thank you for these remarks.**

………………………………………………………………………………………………………………………………………………………………………………………………………………………………

[revised manuscript text omitted]